# Lifelong Learning of Compositional Structures

**Jorge A. Mendez** and **Eric Eaton**
Department of Computer and Information Science
University of Pennsylvania
{mendezme,eeaton}@seas.upenn.edu

## Abstract

A hallmark of human intelligence is the ability to construct self-contained chunks of knowledge and adequately reuse them in novel combinations for solving different yet structurally related problems. Learning such compositional structures has been a significant challenge for artificial systems, due to the combinatorial nature of the underlying search problem. To date, research into compositional learning has largely proceeded separately from work on lifelong or continual learning. We integrate these two lines of work to present a general-purpose framework for lifelong learning of compositional structures that can be used for solving a stream of related tasks. Our framework separates the learning process into two broad stages: learning how to best combine existing components in order to assimilate a novel problem, and learning how to adapt the set of existing components to accommodate the new problem. This separation explicitly handles the trade-off between the stability required to remember how to solve earlier tasks and the flexibility required to solve new tasks, as we show empirically in an extensive evaluation.

## 1 Introduction

A major goal of artificial intelligence is to create an agent capable of acquiring a general understanding of the world. Such an agent would require the ability to continually accumulate and build upon its knowledge as it encounters new experiences. Lifelong machine learning addresses this setting, whereby an agent faces a continual stream of diverse problems and must strive to capture the knowledge necessary for solving each new task it encounters. If the agent is capable of accumulating knowledge in some form of compositional representation (e.g., neural net modules), it could then selectively reuse and combine relevant pieces of knowledge to construct novel solutions.

Various compositional representations for multiple tasks have been proposed recently (Zaremba et al., 2016; Hu et al., 2017; Kirsch et al., 2018; Meyerson & Miikkulainen, 2018). We address the novel question of *how to learn these compositional structures in a lifelong learning setting*. We design a general-purpose framework that is agnostic to the specific algorithms used for learning and the form of the structures being learned. Evoking Piaget's (1976) assimilation and accommodation stages of intellectual development, this framework embodies the benefits of dividing the lifelong learning process into two distinct stages. In the first stage, the learner strives to solve a new task by combining existing components it has already acquired. The second stage uses discoveries from the new task to improve existing components and to construct fresh components if necessary.

Our proposed framework, which we depict visually in Appendix A, is capable of incorporating various forms of compositional structures, as well as different mechanisms for avoiding catastrophic forgetting (McCloskey & Cohen, 1989). As examples of the flexibility of our framework, it can incorporate naïve fine-tuning, experience replay, and elastic weight consolidation (Kirkpatrick et al., 2017) as knowledge retention mechanisms, and linear combinations of linear models (Kumar & Daumé III, 2012; Ruvolo & Eaton, 2013), soft layer ordering (Meyerson & Miikkulainen, 2018), and a soft version of gating networks (Kirsch et al., 2018; Rosenbaum et al., 2018) as the compositional structures. We instantiate our framework with the nine combinations of these examples, and evaluate it on eight different data sets, consistently showing that separating the lifelong learning process into two stages increases the capabilities of the learning system, reducing catastrophic forgetting and achieving higher overall performance. Qualitatively, we show that the components learned by an algorithm that adheres to our framework correspond to self-contained, reusable functions.

## 2  RELATED WORK

**Lifelong learning**    In continual or lifelong learning, agents must handle a variety of tasks over their lifetimes, and should accumulate knowledge in a way that enables them to more efficiently learn to solve new problems. Recent efforts have mainly focused on avoiding catastrophic forgetting. At a high level, algorithms define parts of parametric models (e.g., deep neural networks) to be shared across tasks. As the agent encounters tasks sequentially, it strives to retain the knowledge that enabled it to solve earlier tasks. One common approach is to impose regularization to prevent parameters from deviating in directions that are harmful for performance on the early tasks (Kirkpatrick et al., 2017; Zenke et al., 2017; Li & Hoiem, 2017; Ritter et al., 2018). Another approach retains a small buffer of data from all tasks, and continually updates the model parameters utilizing data from all tasks, thereby maintaining the knowledge required to solve them (Lopez-Paz & Ranzato, 2017; Nguyen et al., 2018; Isele & Cosgun, 2018). A related technique is to learn a generative model to "hallucinate" data, reducing the memory footprint at the cost of using lower-quality data and increasing the cost of accessing data (Shin et al., 2017; Achille et al., 2018; Rao et al., 2019).

These approaches, although effective in avoiding the problem of catastrophic forgetting, make no substantial effort toward the discovery of reusable knowledge. One could argue that the model parameters are learned in such a way that they are reusable across all tasks. However, it is unclear what the reusability of these parameters means, and moreover the way in which parameters are reused is hard-coded into the architecture design. This latter issue is a major drawback when attempting to learn tasks with a high degree of variability, as the exact form in which tasks are related is often unknown. Ideally, the algorithm would be able to determine these relations autonomously.

Other methods learn a set of models that are reusable across many tasks and automatically select how to reuse them (Ruvolo & Eaton, 2013; Nagabandi et al., 2019). However, such methods selectively reuse entire models, enabling knowledge reuse, but not explicitly in a compositional manner.

**Compositional knowledge**    A mostly distinct line of parallel work has explored the learning of compositional knowledge. The majority of such methods either learn the structure for piecing together a given set of components (Cai et al., 2017; Xu et al., 2018; Bunel et al., 2018) or learn the set of components given a known structure for how to compose them (Bošnjak et al., 2017).

A more interesting case is when neither the structure nor the set of components are given, and the agent must autonomously discover the compositional structure underlying a set of tasks. Some approaches for solving this problem assume access to a solution descriptor (e.g., in natural language), which can be mapped by the agent to a solution structure (Hu et al., 2017; Johnson et al., 2017; Pahuja et al., 2019). However, many agents (e.g., service robots) are expected to learn in more autonomous settings, where this kind of supervision is not available. Other approaches instead learn the structure directly from optimization of a cost function (Rosenbaum et al., 2018; Kirsch et al., 2018; Meyerson & Miikkulainen, 2018; Alet et al., 2018; Chang et al., 2019). Many of these works can be viewed as instances of neural architecture search, a closely related area (Elsken et al., 2019).

However, note that the approaches above assume that the agent will have access to a large batch of tasks, enabling it to evaluate numerous combinations of components and structures on all tasks simultaneously. More realistically, the agent faces a sequence of tasks in a lifelong learning fashion. Most work in this line assumes that each component can be fully learned by training on a single task, and then can be reused for other tasks (Reed & de Freitas, 2016; Fernando et al., 2017; Valkov et al., 2018). Unfortunately, this is infeasible in many real-world scenarios in which the agent has access to little data for each of the tasks. One notable exception was proposed by Gaunt et al. (2017), which improves early components with experience in new tasks, but is limited to very simplistic settings.

Unlike prior work, our approach explicitly learns compositional structures in a lifelong learning setting. We do not assume access to a large batch of tasks or the ability to learn definitive components after training on a single task. Instead, we train on a small initial batch of tasks (four tasks, in our experiments), and then autonomously update the existing components to accommodate new tasks.

Our framework also permits incorporating new components over time. Related work has increased network capacity in the non-compositional setting (Yoon et al., 2018) or in a compositional setting where previously learned parameters are kept fixed (Li et al., 2019). Another method enables adaptation of existing parameters (Rajasegaran et al., 2019), but requires expensively storing and training multiple models for each task to select the best one before adapting the existing parameters, and is designed for a specific choice of architecture, unlike our general framework.

## 3 THE LIFELONG LEARNING PROBLEM

We frame lifelong learning as online multi-task learning. The agent will face a sequence of tasks $\mathcal{T}^{(1)}, \ldots, \mathcal{T}^{(T)}$ over its lifetime. Each task will be a learning problem defined by a cost function $\mathcal{L}^{(t)}\big(f^{(t)}\big)$, where the agent must learn a prediction function $f^{(t)} \in \mathcal{F} : \mathcal{X}^{(t)} \mapsto \mathcal{Y}^{(t)}$ to minimize the cost, where $\mathcal{F}$ is a function class, and $\mathcal{X}^{(t)}$ and $\mathcal{Y}^{(t)}$ are the instance and label spaces, respectively. Each task's solution is parameterized by $\boldsymbol{\theta}^{(t)}$, such that $f^{(t)} = f_{\boldsymbol{\theta}^{(t)}}$. The goal of the lifelong learner is to find parameters $\boldsymbol{\theta}^{(1)}, \ldots, \boldsymbol{\theta}^{(T)}$ that minimize the cost across all tasks: $\sum_{t=1}^{T} \mathcal{L}^{(t)}\big(f^{(t)}\big)$. The number of tasks, the order in which tasks will arrive, and the task relationships are all unknown.

Given limited data for each new task, the agent will strive to discover any relevant information to 1) relate it to previously stored knowledge in order to permit transfer and 2) store any new knowledge for future reuse. The agent may be evaluated on any previous task, requiring it to perform well on *all* tasks. In consequence, it must strive to retain knowledge from even the earliest tasks.

## 4 THE LIFELONG COMPOSITIONAL LEARNING FRAMEWORK

Our framework for lifelong learning of compositional structures (illustrated in Appendix A) stores knowledge in a set of $k$ shared components $M = \{m_1, \ldots, m_k\}$ that are acquired and refined over the agent's lifetime. Each component $m_i = m_{\boldsymbol{\phi}_i} \in \mathcal{M}$ is a self-contained, reusable function parameterized by $\boldsymbol{\phi}_i$ that can be combined with other components. The agent reconstructs each task's predictive function $f^{(t)}$ via a task-specific structure $s^{(t)} : \mathcal{X}^{(t)} \times \mathcal{M}^k \mapsto \mathcal{F}$, with $\mathcal{M}^k$ being the set of possible sequences of $k$ components, such that $f^{(t)}(\boldsymbol{x}) = s^{(t)}(\boldsymbol{x}, M)(\boldsymbol{x})$, where $s^{(t)}$ is parameterized by a vector $\boldsymbol{\psi}^{(t)}$. Note that $s^{(t)}$ yields a function from $\mathcal{F}$. The structure functions select the components from $M$ and the order in which to compose them to construct the model for each task (the $f^{(t)}$'s). Specific examples of components and structures are described in Section 4.1.

The intuition behind our framework is that, at any point in time $t$, the agent will have acquired a set of components suitable for solving tasks it encountered previously ($\mathcal{T}^{(1)}, \ldots, \mathcal{T}^{(t-1)}$). If these components, with minor adaptations, can be combined to solve the current task $\mathcal{T}^{(t)}$, then the agent should first learn how to reuse these components before making any modifications to them. The rationale for this idea of keeping components fixed during the early stages of training on the current task $\mathcal{T}^{(t)}$, before the agent has acquired sufficient knowledge to perform well on $\mathcal{T}^{(t)}$, is that premature modification could be catastrophically damaging to the set of existing components. Once the structure $s^{(t)}$ has been learned, we consider that the agent has captured sufficient knowledge about the current task, and it would be sensible to update the components to better accommodate that knowledge. If, instead, it is not possible to capture the current task with the existing components, then new components should be added. These notions loosely mirror the stages of assimilation and accommodation in Piaget's (1976) theories on intellectual development, and so we adopt those terms. Algorithms under our framework take the form of Algorithm 1, split into the following steps.

**Initialization** The components $M$ should be initialized encouraging reusability, both across tasks and within different structural configurations of task models. The former signifies that the components should solve a particular sub-problem regardless of the objective of the task. The latter means that components may be reused multiple times within the structure for a single task's model, or at different structural orders across different tasks. For example, in deep nets, this means that the components could be used at different depths. We achieve this by training the first few tasks the agent encounters jointly to initialize $M$, keeping a fixed, but random, structure that reuses components to encourage reusability.

---

**Algorithm 1** Lifelong Comp. Learning

Initialize components $M$
**while** $\mathcal{T}^{(t)} \leftarrow$ getTask()
    Freeze $M$
    **for** $i = 1, \ldots,$ structUpdates
        Assimilation step on structure $s^{(t)}$
        **if** $i \bmod$ adaptFreq $= 0$
            Freeze $s^{(t)}$, unfreeze $M$
            **for** $j = 1, \ldots,$ compUpdates
                Adaptation step on $M$
            Freeze $M$, unfreeze $s^{(t)}$
    Add components via expansion
    Store info. for future adaptation

---

**Assimilation** Algorithms for finding compositional knowledge vary in how they optimize each task's structure. In modular nets, component selection can be learned via reinforcement learning

(Johnson et al., 2017; Rosenbaum et al., 2018; Chang et al., 2019; Pahuja et al., 2019), stochastic search (Fernando et al., 2017; Alet et al., 2018), or backpropagation (Shazeer et al., 2017; Kirsch et al., 2018; Meyerson & Miikkulainen, 2018). Our framework will use any of these approaches to assimilate the current task by keeping the components $M$ fixed and learning only the structure $s^{(t)}$. Approaches supported by our framework must accept decoupling the learning of the structure from the learning of the components themselves; this requirement holds for all the above examples.

**Accommodation** An effective approach should maintain performance on earlier tasks, while being flexible enough to incorporate new knowledge. To accommodate new knowledge from the current task, the learner may *adapt* existing components or *expand* to include new components:

- *Adaptation step* Approaches for non-compositional structures have been to naïvely fine-tune models with data from the current task, to impose regularization to selectively freeze weights (Kirkpatrick et al., 2017; Ritter et al., 2018), or to store a portion of data from previous tasks and use experience replay (Lopez-Paz & Ranzato, 2017; Isele & Cosgun, 2018). We will instantiate our framework by using any of these methods to accommodate new knowledge into existing components once the current task has been assimilated. For this to be possible, we require that the method can be selectively applied to only the component parameters $\phi$.
- *Expansion step* Often, existing components, even with some adaptation, are insufficient to solve the current task. In this case, the learner would incorporate novel components, which should encode knowledge distinct from existing components and combine with those components to solve the new task. The ability to discover new components endows the learner with the flexibility required to learn over a lifetime. For this, we create *component dropout*, described in Section 4.2.

Concrete instantiations of Algorithm 1 are described in Section 5.1, with pseudocode in Appendix B.

## 4.1 COMPOSITIONAL STRUCTURES

We now present three compositional structures that can be learned within our framework.

**Linear combinations of models** In the simplest setting, each component is a linear model, and they are composed via linear combinations. Specifically, we assume that $\mathcal{X}^{(t)} \subseteq \mathbb{R}^d$, and each task-specific function is given by $f_{\boldsymbol{\theta}^{(t)}}(\boldsymbol{x}) = \boldsymbol{\theta}^{(t)\top}\boldsymbol{x}$, with $\boldsymbol{\theta}^{(t)} \in \mathbb{R}^d$. The predictive functions are constructed from a set of linear component functions $m_{\boldsymbol{\phi}_i}(\boldsymbol{x}) = \boldsymbol{\phi}_i^\top\boldsymbol{x}$, with $\boldsymbol{\phi}_i \in \mathbb{R}^d$, by linearly combining them via a task-specific weight vector $\boldsymbol{\psi}^{(t)} \in \mathbb{R}^k$, yielding: $f^{(t)}(\boldsymbol{x}) = s_{\boldsymbol{\psi}^{(t)}}(\boldsymbol{x}, M)(\boldsymbol{x}) = \boldsymbol{\psi}^{(t)\top}(\boldsymbol{\Phi}^\top\boldsymbol{x})$, where we have constructed the matrix $\boldsymbol{\Phi} = [\boldsymbol{\phi}_1, \ldots, \boldsymbol{\phi}_k]$ to collect all $k$ components.

**Soft layer ordering** In order to handle more complex models, we construct compositional deep nets that compute each layer's output as a linear combination of the outputs of multiple modules. As proposed by Meyerson & Miikkulainen (2018), we assume that each module is one layer, the number of components matches the network's depth, and all components share the input and output dimensions. Concretely, each component is a deep net layer $m_{\boldsymbol{\phi}_i}(\boldsymbol{x}) = \sigma(\boldsymbol{\phi}_i^\top\boldsymbol{x})$, where $\sigma$ is any nonlinear activation and $\boldsymbol{\phi}_i \in \mathbb{R}^{\tilde{d}\times\tilde{d}}$. A set of parameters $\boldsymbol{\psi}^{(t)} \in \mathbb{R}^{k\times k}$ weights the output of the components at each depth: $s^{(t)} = \mathcal{D}^{(t)} \circ \sum_{i=1}^k \boldsymbol{\psi}_{i,1}^{(t)}m_i \circ \cdots \circ \sum_{i=1}^k \boldsymbol{\psi}_{i,k}^{(t)}m_i \circ \mathcal{E}^{(t)}$, where $\mathcal{E}^{(t)}$ and $\mathcal{D}^{(t)}$ are task-specific input and output transformations such that $\mathcal{E}^{(t)} : \mathcal{X}^{(t)} \mapsto \mathbb{R}^{\tilde{d}}$ and $\mathcal{D}^{(t)} : \mathbb{R}^{\tilde{d}} \mapsto \mathcal{Y}^{(t)}$, and the weights are restricted to sum to one at each depth $j$: $\sum_{i=1}^k \boldsymbol{\psi}_{i,j}^{(t)} = 1$.

**Soft gating** In the presence of large data, it is often beneficial to modify the network architecture for each input $\boldsymbol{x}$ (Rosenbaum et al., 2018; Kirsch et al., 2018), unlike both approaches above which use a constant structure for each task. We modify the soft layer ordering architecture by weighting each component's output at depth $j$ by an input-dependent soft gating net $s_j^{(t)} : \mathcal{X}^{(t)} \mapsto \mathbb{R}^k$, giving a predictive function $s^{(t)} = \mathcal{D}^{(t)} \circ \sum_{i=1}^k [s_1^{(t)}(\boldsymbol{x})]_i m_i \circ \cdots \circ \sum_{i=1}^k [s_k^{(t)}(\boldsymbol{x})]_i m_i \circ \mathcal{E}^{(t)}$. As above, we restrict the weights to sum to one at each depth: $\sum_{i=1}^k [s_j^{(t)}(\boldsymbol{x})]_i = 1$.

## 4.2 EXPANSION OF THE SET OF COMPONENTS $M$ VIA COMPONENT DROPOUT

To enable our deep learners to discover new components, we created an expansion step where the agent considers adding a single new component per task. In order to assess the benefit of the new

component, the agent learns two different networks: with and without the novel component. Dropout enables training multiple neural networks without additional storage (Hinton et al., 2012), and has been used to prune neural net nodes in non-compositional settings (Gomez et al., 2019). Our proposed dropout strategy deterministically alternates backpropagation steps with and without the new component, which we call *component dropout*. Intermittently bypassing the new component ensures that existing components can compensate for it if it is discarded. After training, we apply a *post hoc* criterion (in our experiments, a validation error check) to potentially prune the new component.

### 4.3 COMPUTATIONAL COMPLEXITY

Approaches to lifelong learning tend to be computationally intensive, revisiting data or parameters from previous tasks at each training step. Our framework only carries out these expensive operations during (infrequent) adaptation steps. Table 1 summarizes the computational complexity per epoch of the algorithms described in Section 5.1. The assimilation step of our method with expansion (dynamic + compositional) is comparable to compositional baselines in the *worst case* (one new component per task), and our method without expansion (compositional) is always at least as fast.

Table 1: Time complexity per epoch (of assimilation, where applicable) for $n$ samples of $d$ features, $k$ components of $\tilde{d}$ nodes, $T$ tasks, and $n_m$ replay samples per task. Derivations in Appendix C.

| | Dyn. + Comp. | Compositional | Joint | No Comp. |
|---|---|---|---|---|
| ER | | | $O\big((Tn_m+n)\tilde{d}(\tilde{d}k^2+d)\big)$ | $O\big((Tn_m+n)\tilde{d}(\tilde{d}k+d)\big)$ |
| EWC[1] | $O\big(n\tilde{d}(\tilde{d}kT+d)\big)$ | $O\big(n\tilde{d}(\tilde{d}k^2+d)\big)$ | $O\big(n\tilde{d}(T\tilde{d}^2k+\tilde{d}k^2+d)\big)$ | $O\big(n\tilde{d}(T\tilde{d}^2k+\tilde{d}k+d)\big)$ |
| NFT | | | $O\big(n\tilde{d}(\tilde{d}k^2+d)\big)$ | $O\big(n\tilde{d}(\tilde{d}k+d)\big)$ |

## 5 EXPERIMENTAL EVALUATION

### 5.1 FRAMEWORK INSTANTIATIONS AND BASELINES

**Instantiations**  We evaluated our framework with the three compositional structures of Section 4.1. All methods assimilate task $\mathcal{T}^{(t)}$ via backpropagation on the structure's parameters $\psi^{(t)}$. For each, we trained three instantiations of Algorithm 1, varying the method used for adaptation:

- Naïve fine-tuning (**NFT**) updates components via standard backpropagation, ignoring past tasks.
- Elastic weight consolidation (**EWC**, Kirkpatrick et al., 2017) penalizes modifying model parameters via $\frac{\lambda}{2}\sum_{t=1}^{T-1}\|\theta-\theta^{(t)}\|^2_{F^{(t)}}$, where $F^{(t)}$ is the Fisher information around $\theta^{(t)}$. Backpropagation is carried out on the regularized loss, and we approximated $F^{(t)}$ with Kronecker factors.
- Experience replay (**ER**) stores $n_m$ samples per task in a replay buffer, and during adaptation takes backpropagation steps with data from the replay buffer along with the current task's data.

We explored variations with and without the expansion step: **dynamic + compositional** methods use component dropout to add new components, while **compositional** methods keep a fixed-size set.

**Baselines**  For every adaptation method listed above, we constructed two baselines. **Joint** baselines use compositional structures, but do not separate assimilation and accommodation, and instead update components and structures jointly. In contrast, **no-components** baselines optimize a single architecture to be used for all tasks, with additional task-specific input and output mappings, $\mathcal{E}^{(t)}$ and $\mathcal{D}^{(t)}$. The latter baselines correspond to the most common lifelong learning approach, which learns a monolithic structure shared across tasks, while the former are the naïve extensions of those methods to a compositional setting. We also trained an ablated version of our framework that keeps all components fixed after initialization (**FM**), only taking assimilation steps for each new task.

### 5.2 RESULTS

We evaluated these methods on tasks with no evident compositional structure to demonstrate that there is no strict requirement for a certain type of compositionality. Appendix D introduces a simple compositional data set, and shows that our results naturally extend to that setting. We repeated

---

[1]While it is theoretically possible for EWC to operate in constant time w.r.t. $T$, practical implementations use per-task Kronecker factors due to the enormous computational requirements of the constant-time solution.

Table 2: Average final performance across tasks using factored linear models—accuracy for FERA and Landmine (higher is better) and RMSE for Schools (lower is better). Standard errors after $\pm$.

| Base | Algorithm | FERA | Landmine | Schools |
|---|---|---|---|---|
| ER | Compositional | $\mathbf{79.0 \pm 0.4}\%$ | $\mathbf{93.6 \pm 0.1}\%$ | $10.65 \pm 0.04$ |
| | Joint | $78.2 \pm 0.4\%$ | $90.5 \pm 0.3\%$ | $11.55 \pm 0.09$ |
| | No Comp. | $66.4 \pm 0.3\%$ | $93.5 \pm 0.1\%$ | $\mathbf{10.34 \pm 0.02}$ |
| EWC | Compositional | $\mathbf{79.0 \pm 0.4}\%$ | $\mathbf{93.7 \pm 0.1}\%$ | $10.55 \pm 0.03$ |
| | Joint | $72.1 \pm 0.7\%$ | $92.2 \pm 0.2\%$ | $10.73 \pm 0.17$ |
| | No Comp. | $60.1 \pm 0.5\%$ | $93.5 \pm 0.1\%$ | $\mathbf{10.35 \pm 0.02}$ |
| NFT | Compositional | $\mathbf{79.0 \pm 0.4}\%$ | $\mathbf{93.7 \pm 0.1}\%$ | $\mathbf{10.87 \pm 0.07}$ |
| | Joint | $67.9 \pm 0.6\%$ | $72.8 \pm 2.5\%$ | $25.80 \pm 2.35$ |
| | No Comp. | $57.0 \pm 0.9\%$ | $92.7 \pm 0.4\%$ | $18.01 \pm 1.04$ |

experiments ten times with varying random seeds. For details on data sets and hyper-parameters, see Appendix E. Code and data sets are available at `https://github.com/Lifelong-ML/Mendez2020Compositional.git`. Additional results, beyond those presented in this section, are given in Appendix F.

### 5.2.1 LINEAR COMBINATIONS OF MODELS

We first evaluated linear combinations of models on three data sets used previously for evaluating linear lifelong learning (Ruvolo & Eaton, 2013). The Facial Recognition (**FERA**) data set tasks involve recognizing one of three facial expression action units for one of seven people, for a total of $T = 21$ tasks. The **Landmine** data set consists of $T = 29$ tasks, which require detecting land mines in radar images from different regions. Finally, the London Schools (**Schools**) data set contains $T = 139$ regression tasks, each corresponding to exam score prediction in a different school.

Table 2 summarizes the results obtained with linear models. The compositional versions of ER, EWC, and NFT clearly outperformed all the joint versions, which learn the same form of models but by jointly optimizing structures and components. This suggests that the separation of the learning process into assimilation and accommodation stages enables the agent to better capture the structure of the problem. Interestingly, the no-components variants, which learn a single linear model for all tasks, performed better than the jointly trained versions in two out of the three data sets, and even outperformed our compositional algorithms in one. This indicates that the tasks in those two data sets (Landmine and Schools) are so closely related that a single model can capture them.

### 5.2.2 DEEP COMPOSITIONAL LEARNING WITH SOFT LAYER ORDERING

We then evaluated how the different algorithms performed when learning deep nets with soft layer ordering, using five data sets. Binary MNIST (**MNIST**) is a common lifelong learning benchmark, where each task is a binary classification problem between a pair of digits. We constructed $T = 10$ tasks by randomly sampling the digits with replacement across tasks. The Binary Fashion MNIST (**Fashion**) data set is similar to MNIST, but images correspond to items of clothing. For these two data sets, all models used a task-specific input transformation layer $\mathcal{E}^{(t)}$ initialized at random and kept fixed throughout training, to ensure that the input spaces were sufficiently different (Meyerson & Miikkulainen, 2018). A more complex lifelong learning problem commonly used in the literature is Split CUB-200 (**CUB**), where the agent must classify bird species. We created $T = 20$ tasks by randomly sampling ten species for each, without replacement across tasks. All agents used a frozen ResNet-18 pre-trained on ImageNet as a feature extractor $\mathcal{E}^{(t)}$ shared across all tasks. For these first three data sets, all architectures were fully connected networks. To show that our framework supports more complex convolutional architectures, we used two additional data sets. We constructed a lifelong learning version of CIFAR-100 (**CIFAR**) with $T = 20$ tasks by randomly sampling five classes per task, without replacement across tasks. Finally, we used the **Omniglot** data set, which consists of $T = 50$ multi-class classification problems, each corresponding to detecting handwritten symbols in a given alphabet. The inputs to all architectures for these two data sets were the images directly, without any transformation $\mathcal{E}^{(t)}$.

Results in Table 3 show that all the algorithms conforming to our framework outperformed the joint and no-components learners. In four out of the five data sets, the dynamic addition of new components yielded either no or marginal improvements. However, on CIFAR it was crucial for

Table 3: Average final accuracy across tasks using soft layer ordering. Standard errors after $\pm$.

| Base | Algorithm | MNIST | Fashion | CUB | CIFAR | Omniglot |
|------|-----------|-------|---------|-----|-------|----------|
| ER | Dyn. + Comp. | **97.6 ± 0.2**% | **96.6 ± 0.4**% | 79.0 ± 0.5% | **77.6 ± 0.3**% | **71.7 ± 0.5**% |
| | Compositional | 96.5 ± 0.2% | 95.9 ± 0.6% | **80.6 ± 0.3**% | 58.7 ± 0.5% | 71.2 ± 1.0% |
| | Joint | 94.2 ± 0.3% | 95.1 ± 0.7% | 77.7 ± 0.5% | 65.8 ± 0.4% | 70.7 ± 0.3% |
| | No Comp. | 91.2 ± 0.3% | 93.6 ± 0.6% | 44.0 ± 0.9% | 51.6 ± 0.6% | 43.2 ± 4.2% |
| EWC | Dyn. + Comp. | **97.2 ± 0.2**% | **96.5 ± 0.4**% | **73.9 ± 1.0**% | **77.6 ± 0.3**% | **71.5 ± 0.5**% |
| | Compositional | 96.7 ± 0.2% | 95.9 ± 0.6% | 73.6 ± 0.9% | 48.0 ± 1.7% | 53.4 ± 5.2% |
| | Joint | 66.4 ± 1.4% | 69.6 ± 1.6% | 65.4 ± 0.9% | 42.9 ± 0.4% | 58.6 ± 1.1% |
| | No Comp. | 66.0 ± 1.1% | 68.8 ± 1.1% | 50.6 ± 1.2% | 36.0 ± 0.7% | 68.8 ± 0.4% |
| NFT | Dyn. + Comp. | **97.3 ± 0.2**% | **96.4 ± 0.4**% | 73.0 ± 0.7% | **73.0 ± 0.4**% | **69.4 ± 0.4**% |
| | Compositional | 96.5 ± 0.2% | 95.9 ± 0.6% | **74.5 ± 0.7**% | 54.8 ± 1.2% | 68.9 ± 0.9% |
| | Joint | 67.4 ± 1.4% | 69.2 ± 1.9% | 65.1 ± 0.7% | 43.9 ± 0.6% | 63.1 ± 0.9% |
| | No Comp. | 64.4 ± 1.1% | 67.0 ± 1.3% | 49.1 ± 1.6% | 36.6 ± 0.6% | 68.9 ± 1.0% |
| FM | Dyn. + Comp. | **99.1 ± 0.0**% | **97.3 ± 0.3**% | 78.3 ± 0.4% | **78.4 ± 0.3**% | **71.0 ± 0.4**% |
| | Compositional | 84.1 ± 0.8% | 86.3 ± 1.3% | **80.1 ± 0.3**% | 48.8 ± 1.6% | 63.0 ± 3.3% |

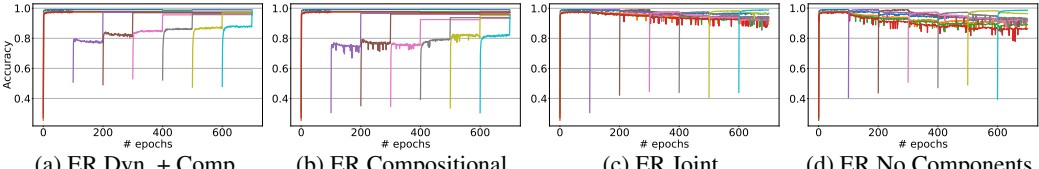

Figure 1: Average gain w.r.t. no-components NFT across tasks and data sets immediately after training on each task (forward) and after all tasks had been trained (final), using soft ordering (top) and soft gating (bottom). Algorithms within our framework (C and D+C) outperformed baselines. Gaps between forward and final performance indicate that our framework exhibits less forgetting.

(a) ER Dyn. + Comp.  (b) ER Compositional  (c) ER Joint  (d) ER No Components

Figure 2: Learning curves averaged across MNIST and Fashion using ER and soft ordering. Each curve shows a single task trained for 100 epochs and continually evaluated during and after training. Algorithms under our framework displayed no forgetting. For ER dynamic + compositional, as more tasks were seen and accommodated, assimilation performance of later tasks improved. Joint and no-components versions dropped performance of early tasks during the learning of later tasks.

the agent to be capable of detecting when new components were needed. This added flexibility enables our learners to handle more varied tasks, where new problems may not be solved without substantially new knowledge. Algorithms with adaptation outperformed the ablated compositional FM agent, showing that it is necessary to accommodate new knowledge into the set of components in order to handle a diversity of tasks. When FM was allowed to dynamically add new components (keeping old ones fixed), it yielded the best performance on MNIST and Fashion by adding far more components than methods with adaptation, as we show in Appendix F, as well as on CIFAR.

To study how flexibly our agents learn new tasks and how stably they retain knowledge about earlier tasks, Figure 1 (top) shows accuracy gains immediately after each task was learned (forward) and after all tasks had been learned (final), w.r.t. no-components NFT (final). Compositional learners with no dynamically added components struggled to match the forward performance of joint baselines, indicating that learning the ordering over existing layers during much of training is less flexible than modifying the layers themselves, as expected. However, the added stability dramatically decreased forgetting w.r.t. joint methods. The dynamic addition of new layers yielded substantial improvements in the forward stage, while still reducing catastrophic forgetting w.r.t. baselines. Figure 2 shows the learning curves of MNIST and Fashion tasks using ER, the best adaptation method. Performance jumps in 100-epoch intervals show adaptation steps incorporating knowledge about the current task into the existing components without noticeably impacting earlier tasks' performance. Compositional and dynamic + compositional ER exhibit almost no performance drop after training

Table 4: Average final accuracy across all tasks using soft gating. Standard errors after the $\pm$.

| Base | Algorithm | MNIST | Fashion | CIFAR | Omniglot |
|------|-----------|-------|---------|-------|----------|
| ER | Dyn. + Comp. | $\mathbf{98.2 \pm 0.1}\%$ | $\mathbf{97.1 \pm 0.4}\%$ | $74.9 \pm 0.3\%$ | $73.7 \pm 0.3\%$ |
| | Compositional | $98.0 \pm 0.2\%$ | $97.0 \pm 0.4\%$ | $\mathbf{75.9 \pm 0.4}\%$ | $\mathbf{73.9 \pm 0.3}\%$ |
| | Joint | $93.8 \pm 0.3\%$ | $94.6 \pm 0.7\%$ | $72.0 \pm 0.4\%$ | $72.6 \pm 0.2\%$ |
| | No Comp. | $91.2 \pm 0.3\%$ | $93.6 \pm 0.6\%$ | $51.6 \pm 0.6\%$ | $43.2 \pm 4.2\%$ |
| EWC | Dyn. + Comp. | $\mathbf{98.2 \pm 0.1}\%$ | $\mathbf{97.0 \pm 0.4}\%$ | $76.6 \pm 0.5\%$ | $73.6 \pm 0.4\%$ |
| | Compositional | $98.0 \pm 0.2\%$ | $\mathbf{97.0 \pm 0.4}\%$ | $\mathbf{76.9 \pm 0.3}\%$ | $\mathbf{74.6 \pm 0.2}\%$ |
| | Joint | $68.6 \pm 0.9\%$ | $69.5 \pm 1.8\%$ | $49.9 \pm 1.1\%$ | $63.5 \pm 1.2\%$ |
| | No Comp. | $66.0 \pm 1.1\%$ | $68.8 \pm 1.1\%$ | $36.0 \pm 0.7\%$ | $68.8 \pm 0.4\%$ |
| NFT | Dyn. + Comp. | $\mathbf{98.2 \pm 0.1}\%$ | $\mathbf{97.1 \pm 0.4}\%$ | $66.6 \pm 0.7\%$ | $69.1 \pm 0.9\%$ |
| | Compositional | $98.0 \pm 0.2\%$ | $96.9 \pm 0.5\%$ | $\mathbf{68.2 \pm 0.5}\%$ | $\mathbf{72.1 \pm 0.3}\%$ |
| | Joint | $67.3 \pm 1.7\%$ | $66.4 \pm 1.9\%$ | $51.0 \pm 0.8\%$ | $65.8 \pm 1.3\%$ |
| | No Comp. | $64.4 \pm 1.1\%$ | $67.0 \pm 1.3\%$ | $36.6 \pm 0.6\%$ | $68.9 \pm 1.0\%$ |
| FM | Dyn. + Comp. | $\mathbf{98.4 \pm 0.1}\%$ | $\mathbf{97.0 \pm 0.4}\%$ | $\mathbf{77.2 \pm 0.3}\%$ | $74.0 \pm 0.4\%$ |
| | Compositional | $94.8 \pm 0.4\%$ | $96.3 \pm 0.4\%$ | $\mathbf{77.2 \pm 0.3}\%$ | $\mathbf{74.1 \pm 0.3}\%$ |

Table 5: Average final accuracy across tasks on the Combined data set. Each column shows accuracy on the subset of tasks from each given data set, as labeled. Standard errors after $\pm$.

| Base | Algorithm | All data sets | MNIST | Fashion | CUB |
|------|-----------|---------------|-------|---------|-----|
| ER | Dyn. + Comp. | $\mathbf{86.5 \pm 1.8}\%$ | $\mathbf{99.5 \pm 0.0}\%$ | $\mathbf{98.0 \pm 0.3}\%$ | $\mathbf{74.2 \pm 2.0}\%$ |
| | Compositional | $82.1 \pm 2.5\%$ | $\mathbf{99.5 \pm 0.0}\%$ | $97.8 \pm 0.3\%$ | $65.5 \pm 2.4\%$ |
| | Joint | $72.8 \pm 4.1\%$ | $98.9 \pm 0.3\%$ | $97.0 \pm 0.7\%$ | $47.6 \pm 6.2\%$ |
| | No Comp. | $47.4 \pm 4.5\%$ | $91.8 \pm 1.3\%$ | $83.5 \pm 2.5\%$ | $7.1 \pm 0.4\%$ |

on a task, whereas accuracy for the joint and no-components versions diminishes as the agent learns subsequent tasks. Most notably, as more tasks were seen by dynamic ER, the existing components became better able to assimilate new tasks, shown by the trend of increasing performance as the number of tasks increases. This suggests that the later tasks' accommodation stage can successfully determine which new knowledge should be incorporated into existing components (enabling them to better generalize across tasks), and which must be incorporated into a new component.

In Appendix F, we show that our methods do not forget early tasks, and outperform baselines even in small data settings. We also found that most components learned by our methods are reused by multiple tasks, as desired. Moreover, analysis of various accommodation schedules revealed that infrequent adaptation leads to best results, informing future algorithm design choices.

### 5.2.3 DEEP COMPOSITIONAL LEARNING WITH SOFT GATING

Finally, we tested our algorithms when the compositional structures were given by a soft gating net. Table 4 shows a substantial improvement of compositional algorithms w.r.t. baselines. We hypothesized that the gating net granted our assimilation step more flexibility, which is confirmed in Figure 1 (bottom): the forward accuracy of compositional methods was nearly identical to that of jointly trained and no-components versions. This added flexibility enabled our simplest version of a compositional algorithm, FM, to perform better than the full versions of our algorithm on the CIFAR data set with convolutional gating nets, showing that even the components initialized with only a few tasks are sufficient for top lifelong learning performance. We attempted to run experiments with this method on the CUB data set, but found that all algorithms were incapable of generalizing to test data. This is consistent with findings in prior work, which showed that gating nets require vast amounts of data, unavailable in CUB (Rosenbaum et al., 2018; Kirsch et al., 2018).

### 5.2.4 DEEP COMPOSITIONAL LEARNING OF SEQUENCES OF DIVERSE TASKS

One of the key advantages of learning compositional structures is that they enable learning a more diverse set of tasks, by recombining components in novel ways to solve each problem. In this setting, non-compositional structures struggle to capture the diversity of the tasks in a single monolithic architecture. To verify that this is indeed the case, we created a novel data set that combines the MNIST, Fashion, and CUB data sets into a single **Combined** data set of $T = 40$ tasks. We trained all our instantiations and baselines with soft layer ordering, using the same architecture as used for CUB in Section 5.2.2. Agents were given no indication that each task came from a different data

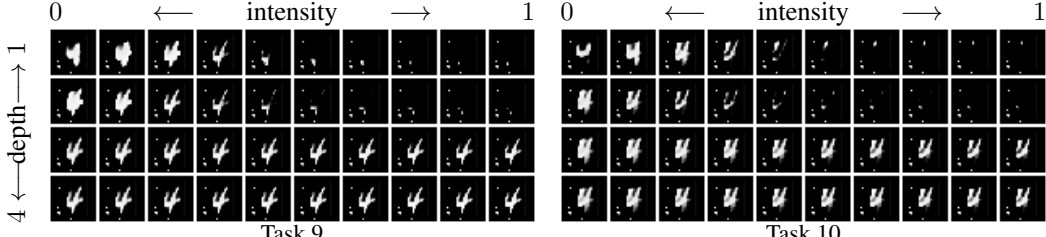

Figure 3: Generated MNIST "4" digits on the last two tasks seen by compositional ER with soft ordering, varying the intensity with which a single specific component is selected. The learned component performs a functional primitive: the more the component is used (left to right on each row), the thinner the lines of the digit become. The magnitude of this effect decreases with depth (moving from top to bottom), with the digit completely disappearing at the earliest layers, but only becoming slightly sharper at the deepest layers. This effect is consistent across both tasks.

set, and they were all trained following the exact same setup of Section 5.2.2. Table 5 summarizes the results for ER-based algorithms, showing that our method greatly outperforms the baselines. In particular, ER with no components was completely incapable of learning the CUB tasks, showing that compositional architectures are required to handle this more complex setting. Even jointly training the components and structures performed poorly. Algorithms under our framework instead performed remarkably well, especially the complete version with dynamically added components. The remaining instantiations and baselines exhibited a similar behavior (see Appendix G).

### 5.3 VISUALIZATION OF THE LEARNED COMPONENTS

We now visually inspect the components learned by our framework to verify that they are indeed self-contained and reusable. Similar to Meyerson & Miikkulainen (2018), we created $T = 10$ generative tasks, where each pixel in an image of the digit "4" constitutes one data point, using the coordinates as features and the pixel intensity as the label. We trained a soft ordering net with $k = 4$ components via compositional ER, and shared the input and output transformations across tasks to ensure the only differences across task models are due to the structure $s^{(t)}$ of each task. Varying the intensity $\psi_{i,j}^{(t)}$ with which component $i$ is selected at various depths $j$ gives information about the effect of the component in different contexts. Figure 3 shows generated digits as the intensity of component $i = 0$ varies at different depths, revealing that the discovered component learned to vary the thickness of the digit regardless of the task at hand, with a more extreme effect at the initial layers. Additional details and more comprehensive visualizations are included in Appendix H.

### CONCLUSION

We presented a general framework for learning compositional structures in a lifelong learning setting. The key piece of our framework is the separation of the learning process into two stages: assimilation of new problems with existing components, and accommodation of newly discovered knowledge into the set of components. These stages have connections to Piagetian theory of development, opening the door for future investigations that bridge between lifelong machine learning and developmental psychology. We showed the flexibility of our framework by capturing nine different concrete algorithms within our framework, and demonstrated empirically in an extensive evaluation that these algorithms are stronger lifelong learners than existing approaches. More concretely, we demonstrated that both learning traditional monolithic architectures and naïvely training compositional structures via existing methods lead to substantially degraded performance. Our framework is simple conceptually, easy to combine with existing continual or compositional learning techniques, and effective in trading off the flexibility and stability required for lifelong learning.

In this work, we showed the potential of compositional structures to enable strong lifelong learning. One major line of open work remains properly understanding how to measure the quality of the obtained compositional solutions, especially in settings without obvious decompositions, like those we consider in Section 5.2. While our visualizations in Section 5.3 and results in Table F.4 suggest that our method obtains reusable components, we currently lack a proper metric to assess the degree to which the learned structures are compositional. We leave this question open for future investigation.

ACKNOWLEDGMENTS

We would like to thank Seungwon Lee, Boyu Wang, and Oleh Rybkin for their feedback on various versions of this draft. We would also like to thank the anonymous reviewers for their valuable feedback and suggestions. The research presented in this paper was partially supported by the DARPA Lifelong Learning Machines program under grant FA8750-18-2-0117, the DARPA SAIL-ON program under contract HR001120C0040, and the Army Research Office under MURI grant W911NF-20-1-0080. The views and conclusions in this paper are those of the authors and should not be interpreted as representing the official policies, either expressed or implied, of the Defense Advanced Research Projects Agency, the Army Research Office, or the U.S. Government. The U.S. Government is authorized to reproduce and distribute reprints for Government purposes notwithstanding any copyright notation herein.

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

# APPENDICES TO
# "LIFELONG LEARNING OF COMPOSITIONAL STRUCTURES"

### by **Jorge A. Mendez** and **Eric Eaton**

## A    DEPICTION OF OUR COMPOSITIONAL LIFELONG LEARNING FRAMEWORK

Section 4 in the main paper presented our general-purpose framework for lifelong learning of compositional structures. Figure A.1 illustrates our proposed framework, split into four learning stages: 1) initialization of components, 2) assimilation of new tasks with existing components, 3) adaptation of existing components with new knowledge, and 4) expansion of the set of components.

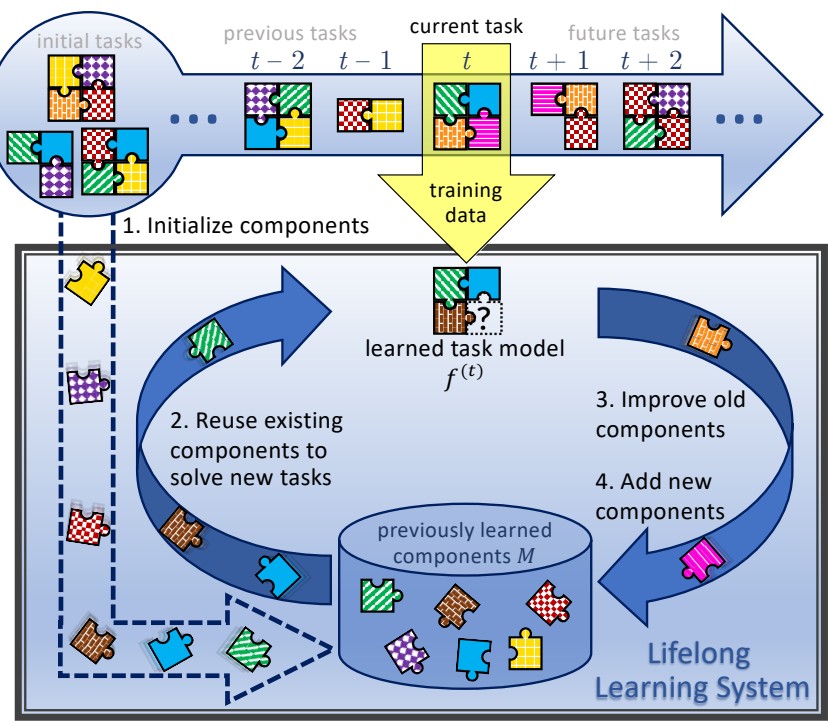

Figure A.1: Our framework for lifelong compositional learning initializes a set of components by training on a small set of tasks (1). Each new task is learned by composing the relevant components (2). Subsequently, the agent improves imperfect components with newly discovered knowledge (3), and adds any new components that were discovered during training on the current task (4).

## B    FRAMEWORK INSTANTIATIONS

In this section, we describe in more detail the specific implementations used in the experiments of Section 5 in the main paper. To simplify evaluation, we fixed `structUpdates`, `adaptFreq`, and `compUpdates` such that the learning process would be split into multiple epochs of assimilation followed by a single final epoch of adaptation. This is the most extreme separation of the learning into assimilation and accommodation: no knowledge is accommodated into existing components until after assimilation has finished. We study the effects of this choice in Appendix F.

Algorithms B.5–B.10 summarize the implementations of all instantiations of our framework used in our experiments. Shared subroutines are included as Algorithms B.1–B.4, and we leave blank lines in compositional methods to highlight missing steps from their dynamic + compositional counter-

parts. The learner first initializes the components by jointly training on the first $T_{\texttt{init}}$ tasks it encounters. At every subsequent time $t$, during the first $(\texttt{numEpochs} - 1)$ epochs, the agent assimilates the new task by training the task-specific structure parameters $\boldsymbol{\psi}^{(t)}$ via backpropagation, learning how to combine existing components for task $\mathcal{T}^{(t)}$. For dynamic + compositional methods, this assimilation step is done with component dropout, and simultaneously optimizes the parameters of the newly added component $\boldsymbol{\phi}_{k+1}$. The adaptation step varies according to the base lifelong learning method, applying techniques for avoiding forgetting to the whole set of component parameters $\boldsymbol{\Phi}$ for one epoch. This step is done via component dropout for dynamic + compositional methods. Finally, dynamic + compositional methods discard the fresh component if it does not improve performance by more than a threshold $\tau$ on the current task $\mathcal{T}^{(t)}$, and otherwise keep it for future training.

---

**Algorithm B.1** Initialization

1: $s^{(t)} \leftarrow \texttt{randomInitialization}()$
2: $\texttt{init\_buff} \leftarrow \texttt{init\_buff} \cup \mathcal{T}^{(t)}.\texttt{train}$
3: **if** $t = T_{\texttt{init}} - 1$
4:    **for** $i = 1, \dots, \texttt{numEpochs}$
5:       **for** $\tilde{t}, \boldsymbol{x} \leftarrow \texttt{init\_buff}$
6:          $\boldsymbol{\Phi} \leftarrow \boldsymbol{\Phi} - \eta \nabla_{\boldsymbol{\Phi}} \mathcal{L}^{(\tilde{t})}(f^{(\tilde{t})}(\boldsymbol{x}))$
7:       **end for**
8:    **end for**    ▷ backprop on components
9: **end if**

---

**Algorithm B.2** Expansion

1: $a_1 \leftarrow \texttt{accuracy}(\mathcal{T}^{(t)}.\texttt{validation})$
2: $\texttt{hideComponent}(k + 1)$
3: $a_2 \leftarrow \texttt{accuracy}(\mathcal{T}^{(t)}.\texttt{validation})$
4: $\texttt{recoverComponent}(k + 1)$
5: **if** $\frac{a_1 - a_2}{a_2} < \tau$    ▷ validation error check
6:    $\texttt{discardComponent}(k + 1)$
7:    $k \leftarrow k + 1$
8: **end if**

---

**Algorithm B.3** Assimilation (Comp.)

1: **for** $i = 1, \dots, \texttt{numEpochs} - 1$
2:    **for** $\boldsymbol{x} \leftarrow \mathcal{T}^{(t)}.\texttt{train}$
3:       $\boldsymbol{\psi}^{(t)} \leftarrow \boldsymbol{\psi}^{(t)} - \eta \nabla_{\boldsymbol{\psi}^{(t)}} \mathcal{L}^{(t)}(f^{(t)}(\boldsymbol{x}))$

4:    **end for**
5: **end for**    ▷ backprop on structure

---

**Algorithm B.4** Assimilation (Dyn. + Comp.)

1: $\boldsymbol{\Phi} \leftarrow [\boldsymbol{\Phi}; \texttt{randomVector}()]$   ▷ new comp.
2: $\boldsymbol{\psi}^{(t)}_{k+1, 1:k} \leftarrow 1$
3: **for** $i = 1, \dots, \texttt{numEpochs} - 1$
4:    **for** $\boldsymbol{x} \leftarrow \mathcal{T}^{(t)}.\texttt{train}$
5:       $\boldsymbol{\psi}^{(t)} \leftarrow \boldsymbol{\psi}^{(t)} - \eta \nabla_{\boldsymbol{\psi}^{(t)}} \mathcal{L}^{(t)}(f^{(t)}(\boldsymbol{x}))$
6:       $\boldsymbol{\phi}_{k+1} \leftarrow \boldsymbol{\phi}_{k+1} - \eta \nabla_{\boldsymbol{\phi}_{k+1}} \mathcal{L}^{(t)}(f^{(t)}(\boldsymbol{x}))$
7:       $\texttt{hideComponent}(k + 1)$
8:       $\boldsymbol{\psi}^{(t)} \leftarrow \boldsymbol{\psi}^{(t)} - \eta \nabla_{\boldsymbol{\psi}^{(t)}} \mathcal{L}^{(t)}(f^{(t)}(\boldsymbol{x}))$
9:       $\texttt{recoverComponent}(k + 1)$
10:   **end for**
11: **end for**    ▷ component dropout

---

**Algorithm B.5** Compositional ER

1: **while** $\mathcal{T}^{(t)} \leftarrow \texttt{getTask}()$
2:    **if** $t < T_{\texttt{init}}$
3:       Call Algorithm B.1    ▷ initialization
4:    **else**
5:       Call Algorithm B.3    ▷ assimilation
6:       **for** $\tilde{t}, \boldsymbol{x} \leftarrow (t, \mathcal{T}^{(t)}.\texttt{train}) \cup \texttt{buffer}$
7:          $\boldsymbol{\Phi} \leftarrow \boldsymbol{\Phi} - \eta \nabla_{\boldsymbol{\Phi}} \mathcal{L}^{(\tilde{t})}(f^{(\tilde{t})}(\boldsymbol{x}))$

8:       **end for**    ▷ adaptation

9:    **end if**
10:   $\texttt{buffer}[t] \leftarrow \texttt{sample}(\mathcal{T}^{(t)}.\texttt{train}, n_m)$
11: **end while**

---

**Algorithm B.6** Dynamic + Compositional ER

1: **while** $\mathcal{T}^{(t)} \leftarrow \texttt{getTask}()$
2:    **if** $t < T_{\texttt{init}}$
3:       Call Algorithm B.1    ▷ initialization
4:    **else**
5:       Call Algorithm B.4    ▷ assimilation
6:       **for** $\tilde{t}, \boldsymbol{x} \leftarrow (t, \mathcal{T}^{(t)}.\texttt{train}) \cup \texttt{buffer}$
7:          $\boldsymbol{\Phi} \leftarrow \boldsymbol{\Phi} - \eta \nabla_{\boldsymbol{\Phi}} \mathcal{L}^{(\tilde{t})}(f^{(\tilde{t})}(\boldsymbol{x}))$
8:          $\texttt{hideComponent}(k + 1)$
9:          $\boldsymbol{\Phi} \leftarrow \boldsymbol{\Phi} - \eta \nabla_{\boldsymbol{\Phi}} \mathcal{L}^{(\tilde{t})}(f^{(\tilde{t})}(\boldsymbol{x}))$
10:      $\texttt{recoverComponent}(k + 1)$
11:      **end for**    ▷ adaptation
12:      Call Algorithm B.2    ▷ expansion
13:    **end if**
14:   $\texttt{buffer}[t] \leftarrow \texttt{sample}(\mathcal{T}^{(t)}.\texttt{train}, n_m)$
15: **end while**

---

**Algorithm B.7** Compositional EWC

1: **while** $\mathcal{T}^{(t)} \leftarrow \texttt{getTask}()$
2:    **if** $t < T_{\texttt{init}}$
3:       Call Algorithm B.1    ▷ initialization
4:    **else**
5:       Call Algorithm B.3    ▷ assimilation
6:       **for** $\boldsymbol{x} \leftarrow \mathcal{T}^{(t)}.\texttt{train}$
7:          $\boldsymbol{A} \leftarrow \sum_{\tilde{t}}^{t-1} \boldsymbol{a}^{(\tilde{t})} \boldsymbol{\Phi} \boldsymbol{b}^{(\tilde{t})}$
8:          $\boldsymbol{g} \leftarrow \nabla_{\boldsymbol{\Phi}} \mathcal{L}^{(t)}(f^{(t)}(\boldsymbol{x})) + \lambda(\boldsymbol{A} - \boldsymbol{B})$
9:          $\boldsymbol{\Phi} \leftarrow \boldsymbol{\Phi} - \eta \boldsymbol{g}$
10:       **end for**       ▷ adaptation
11:    **end if**
12:    $\boldsymbol{a}^{(t)}, \boldsymbol{b}^{(t)} \leftarrow \texttt{KFAC}(\mathcal{T}^{(t)}.\texttt{train}, \boldsymbol{\Phi})$
13:    $\boldsymbol{B} \leftarrow \boldsymbol{B} - \boldsymbol{a}^{(t)} \boldsymbol{\Phi} \boldsymbol{b}^{(t)}$
14: **end while**

---

**Algorithm B.8** Dynamic + Compositional EWC

1: **while** $\mathcal{T}^{(t)} \leftarrow \texttt{getTask}()$
2:    **if** $t < T_{\texttt{init}}$
3:       Call Algorithm B.1    ▷ initialization
4:    **else**
5:       Call Algorithm B.4    ▷ assimilation
6:       **for** $\boldsymbol{x} \leftarrow \mathcal{T}^{(t)}.\texttt{train}$
7:          $\boldsymbol{A} \leftarrow \sum_{\tilde{t}}^{t-1} \boldsymbol{a}^{(\tilde{t})} \boldsymbol{\Phi} \boldsymbol{b}^{(\tilde{t})}$
8:          $\boldsymbol{g} \leftarrow \nabla_{\boldsymbol{\Phi}} \mathcal{L}^{(t)}(f^{(t)}(\boldsymbol{x})) + \lambda(\boldsymbol{A} - \boldsymbol{B})$
9:          $\boldsymbol{\Phi} \leftarrow \boldsymbol{\Phi} - \eta \boldsymbol{g}$
10:          $\texttt{hideComponent}(k+1)$
11:          $\boldsymbol{A} \leftarrow \sum_{\tilde{t}}^{t-1} \boldsymbol{a}^{(\tilde{t})} \boldsymbol{\Phi} \boldsymbol{b}^{(\tilde{t})}$
12:          $\boldsymbol{g} \leftarrow \nabla_{\boldsymbol{\Phi}} \mathcal{L}^{(t)}(f^{(t)}(\boldsymbol{x})) + \lambda(\boldsymbol{A} - \boldsymbol{B})$
13:          $\boldsymbol{\Phi} \leftarrow \boldsymbol{\Phi} - \eta \boldsymbol{g}$
14:          $\texttt{recoverComponent}(k+1)$
15:       **end for**       ▷ adaptation
16:       Call Algorithm B.2    ▷ expansion
17:    **end if**
18:    $\boldsymbol{a}^{(t)}, \boldsymbol{b}^{(t)} \leftarrow \texttt{KFAC}(\mathcal{T}^{(t)}.\texttt{train}, \boldsymbol{\Phi})$
19:    $\boldsymbol{B} \leftarrow \boldsymbol{B} - \boldsymbol{a}^{(t)} \boldsymbol{\Phi} \boldsymbol{b}^{(t)}$
20: **end while**

---

**Algorithm B.9** Compositional NFT

1: **while** $\mathcal{T}^{(t)} \leftarrow \texttt{getTask}()$
2:    **if** $t < T_{\texttt{init}}$
3:       Call Algorithm B.1    ▷ initialization
4:    **else**
5:       Call Algorithm B.3    ▷ assimilation
6:       **for** $\boldsymbol{x} \leftarrow \mathcal{T}^{(t)}.\texttt{train}$
7:          $\boldsymbol{\Phi} \leftarrow \boldsymbol{\Phi} - \eta \nabla_{\boldsymbol{\Phi}} \mathcal{L}^{(t)}(f^{(t)}(\boldsymbol{x}))$
8:       **end for**       ▷ adaptation
9:    **end if**
10: **end while**

---

**Algorithm B.10** Dynamic + Compositional NFT

1: **while** $\mathcal{T}^{(t)} \leftarrow \texttt{getTask}()$
2:    **if** $t < T_{\texttt{init}}$
3:       Call Algorithm B.1    ▷ initialization
4:    **else**
5:       Call Algorithm B.4    ▷ assimilation
6:       **for** $\boldsymbol{x} \leftarrow \mathcal{T}^{(t)}.\texttt{train}$
7:          $\boldsymbol{\Phi} \leftarrow \boldsymbol{\Phi} - \eta \nabla_{\boldsymbol{\Phi}} \mathcal{L}^{(t)}(f^{(t)}(\boldsymbol{x}))$
8:          $\texttt{hideComponent}(k+1)$
9:          $\boldsymbol{\Phi} \leftarrow \boldsymbol{\Phi} - \eta \nabla_{\boldsymbol{\Phi}} \mathcal{L}^{(t)}(f^{(t)}(\boldsymbol{x}))$
10:          $\texttt{recoverComponent}(k+1)$
11:       **end for**       ▷ adaptation
12:       Call Algorithm B.2    ▷ expansion
13:    **end if**
14: **end while**

---

## C   COMPUTATIONAL COMPLEXITY DERIVATION

We now derive asymptotic bounds for the computational complexity of all baselines and instantiations of our framework presented in Section 4.3 in the main paper. We assume the network architecture uses fully connected layers, and soft layer ordering for compositional structures. Extending these results to convolutional layers and soft gating is straightforward.

A single forward and backward pass through a standard fully connected layer of $i$ inputs and $o$ outputs requires $O(io)$ computations, and is additive across layers. Assuming a binary classification net, the no-components architecture contains one input layer $\mathcal{E}^{(t)}$ with $d$ inputs and $\tilde{d}$ outputs, $k$ layers with $\tilde{d}$ inputs and $\tilde{d}$ outputs, and one output layer $\mathcal{D}^{(t)}$ with $\tilde{d}$ inputs and one output. Training such a net in the standard single-task setting then requires $O\big(d\tilde{d} + \tilde{d}^2 k + \tilde{d}\big)$ computations per input point. For a full epoch of training on a data set with $n$ data points, the training cost would then be $O\big(n\tilde{d}(\tilde{d}k + d)\big)$. This is exactly the computational cost of no-components NFT, since it ignores any information from past tasks during training, and leverages only the initialization of parameters.

On the other hand, a soft layer ordering net evaluates all $k$ layers of size $\tilde{d} \times \tilde{d}$ at every one of the $k$ depths in the network, resulting in a cost of $O\big(\tilde{d}^2 k^2\big)$ for those layers. This results in an overall cost per epoch of $O\big(n\tilde{d}\big(\tilde{d}k^2 + d\big)\big)$ for single-task training, and therefore also for joint NFT training. Since compositional methods do not use information from earlier tasks during assimilation, because they only train the task-specific structure $s^{(t)}$ during this stage, then the cost per epoch of assimilation is also $O\big(n\tilde{d}\big(\tilde{d}k^2 + d\big)\big)$. Dynamic + compositional methods can at most contain $T$ components if they add one new component for every seen task. This leads to a cost of $O\big(\tilde{d}^2 kT\big)$ for the shared layers, and an overall cost per epoch of assimilation of $O\big(n\tilde{d}\big(\tilde{d}kT + d\big)\big)$.

Kronecker-factored EWC requires computing two $O\big(\tilde{d} \times \tilde{d}\big)$ matrices, $\boldsymbol{a}^{(t)}$ and $\boldsymbol{b}^{(t)}$, for every observed task. At each training iteration, EWC modifies the gradient of component $i$ by adding $\lambda \sum_{t=1}^{T} \boldsymbol{a}^{(t)} \boldsymbol{\phi}_i \boldsymbol{b}^{(t)} - \boldsymbol{a}^{(t)} \boldsymbol{\phi}_i^{(t)} \boldsymbol{b}^{(t)}$, where $\boldsymbol{\phi}_i^{(t)}$ are the parameters of component $i$ obtained after training on task $\mathcal{T}^{(t)}$. While the second term of this sum can be pre-computed and stored in memory, it is not possible to pre-compute the first term. Theoretically, one can apply Kronecker product properties to store a (prohibitively large) $O\big(\tilde{d}^2 \times \tilde{d}^2\big)$ matrix and avoid computing the per-task sum, but practical implementations avoid this and instead compute the sum for every task, at a cost of $O\big(T\tilde{d}^3 k\big)$ per mini-batch. With $O(n)$ mini-batches per epoch, we obtain an additional cost with respect to joint and no-components NFT of $O\big(nT\tilde{d}^3 k\big)$. Note that this step is carried out after obtaining the gradients for each layer, and thus there is no additional $k^2$ term for joint EWC.

Deriving the complexity bound of ER simply requires extending the size of the batch of data from $n$ to $(Tn_m + n)$ for a replay buffer size of $n_m$ per task.

To put the computational complexity of dynamic + compositional methods into perspective, we compute the number of components required to solve $T$ tasks. We consider networks with *hard* layer ordering, and assume that all $T$ tasks can be represented by different orders over the same set of components. Given a network with $k$ depths and $\tilde{k}$ components, it is possible to create $\tilde{k}^k$ different layer orderings. If all $T$ tasks require different orderings, then we require at least $\tilde{k} = \sqrt[k]{T}$ components. Designing a lifelong learning algorithm that can provably attain this bound in the number of components, or any sublinear growth in $T$, remains an open problem.

For completeness, we note that the (very infrequent) adaptation steps for compositional methods incur the same computational cost as any epoch of joint methods. On the other hand, to obtain the cost of adaptation steps for dynamic + compositional methods, we need to replace $k^2$ terms in the expressions for joint methods by $kT$, again noting that this corresponds to the *worst case*, where the agent adds a new component for every single task it encounters.

## D    EVALUATION ON A TOY COMPOSITIONAL DATA SET

The results of Section 5.2 in the main paper were obtained on a suite of data sets that does not explicitly require any compositional structure. This deliberate choice enabled us to study the generality of our framework, and we found that algorithms that instantiate it work well across data sets with a range of feature representations, relations across tasks, number of tasks, and sample sizes. In this section, we introduce a data set that explicitly assumes a compositional structure that intuitively matches the assumptions of our soft layer ordering architecture, and we show that the results obtained for non-compositional data sets still hold for this class of problems.

We created the **Objects** data set with 48 classes, each corresponding to an object composed of a shape (circle, triangle, or square), color (orange, blue, pink, or green), and location (each of the for quadrants in the image). We generated $n = 100$ images of size $28 \times 28$ per class. We uniformly sampled the center of the object from $[c_x - 3, c_x + 3], [c_y - 3, c_y + 3]$, where $c_x$ and $c_y$ are the centers of the quadrant for each class, respectively. The RGB values were uniformly sampled from $[r - 16, r + 16], [g - 16, g + 16], [b - 16, g + 16]$, where $r$, $g$, and $b$ are the nominal RGB values for the color of the class. Finally, we uniformly sampled the size of the objects from $[3, 7]$ pixels.

To test our approach in this setting, we created a lifelong version of the Objects data set by randomly splitting the data into 16 three-way classification tasks. $50\%$ of the instances for each class were used as training data, $20\%$ as validation data, and $30\%$ as test data. We used soft ordering nets with $k = 4$

Table D.1: Average final accuracy across tasks on the compositional Objects data set using soft layer ordering. Column labels indicate which component was held out for final tasks. Std. errors after $\pm$.

| Base | Algorithm | Circle | Top-left | Orange | Random |
|------|-----------|--------|----------|--------|--------|
| ER | Dyn. + Comp. | **93.4 ± 0.7**% | **85.9 ± 1.0**% | **89.4 ± 1.0**% | **91.8 ± 0.6**% |
| | Compositional | 92.2 ± 0.9% | 84.9 ± 1.2% | 88.7 ± 1.2% | 90.9 ± 0.9% |
| | Joint | 92.0 ± 0.8% | 83.5 ± 1.0% | 87.8 ± 1.2% | 89.1 ± 0.6% |
| | No Comp. | 91.2 ± 1.0% | 83.5 ± 1.1% | 88.4 ± 0.8% | 89.8 ± 1.0% |
| EWC | Dyn. + Comp. | **93.4 ± 0.7**% | **85.9 ± 1.0**% | **89.5 ± 1.1**% | **91.6 ± 0.7**% |
| | Compositional | 92.0 ± 1.1% | 85.3 ± 1.2% | 88.7 ± 1.2% | 90.9 ± 0.9% |
| | Joint | 91.1 ± 0.8% | 82.4 ± 1.3% | 87.0 ± 1.4% | 90.1 ± 0.7% |
| | No Comp. | 88.1 ± 1.8% | 81.0 ± 1.5% | 83.3 ± 2.5% | 86.3 ± 2.1% |
| NFT | Dyn. + Comp. | **93.3 ± 0.7**% | **86.3 ± 1.0**% | **89.6 ± 1.1**% | **91.5 ± 0.6**% |
| | Compositional | 92.3 ± 0.8% | 85.7 ± 1.1% | 88.7 ± 1.2% | 90.6 ± 0.9% |
| | Joint | 90.6 ± 0.9% | 81.8 ± 1.2% | 86.6 ± 1.3% | 88.4 ± 1.2% |
| | No Comp. | 89.2 ± 2.0% | 77.5 ± 1.9% | 86.8 ± 1.2% | 85.7 ± 1.5% |
| FM | Dyn. + Comp. | **93.0 ± 0.7**% | **86.0 ± 1.0**% | **89.5 ± 1.1**% | **91.4 ± 0.5**% |
| | Compositional | 90.8 ± 1.4% | 83.8 ± 1.5% | 88.1 ± 1.2% | 89.4 ± 0.9% |

components of 64 fully connected hidden units shared across tasks, and a linear input transformation $\mathcal{E}^{(t)}$ trained for each task. All agents trained for 100 epochs per task using a mini-batch of size 32, with compositional agents using 99 epochs for assimilation and a single epoch for adaptation. The regularization hyper-parameter for EWC was set to $\lambda = 1e - 3$, and ER was given a a replay buffer of size $n_m = 5$. We ran 50 trials of each experiment with different random seeds controlling class splits for each task, training/validation/test splits for each class, and the ordering of tasks.

We evaluated all methods in four different settings. In the Random setting, classes were randomly split and ordered, matching the experimental setting of Section 5.2. The other, more challenging settings were created by holding out one shape, location, or color only for the final four (for color and location) or five (for shape) tasks, requiring the agents to adapt to never-seen components dynamically. Results in Table D.1 show each of our methods outperforms all baselines in all settings, showcasing the ability of our framework to discover the underlying compositional structures.

# E EXPERIMENTAL SETTING

Below, we give additional details describing the experimental setting used in the main paper.

## E.1 DATA SETS

The data sets used for linear experiments underwent the same processing and train/test split of Ruvolo & Eaton (2013). For MNIST and Fashion, we randomly sampled pairs of digits to act as the positive and negative classes in each task, and allowed digits to be reused across tasks. For CUB and CIFAR, ten and five classes were used per task, respectively, without reusing of classes across different tasks. CUB images were cropped by the provided bounding boxes and resized to $224 \times 224$. For these four data sets, we used the standard train/test split, and further divided the training set into 80% for training and 20% for validation. Finally, for Omniglot, we used each alphabet as one task, and split the data into 80% for training, 10% for validation, and 10% for test, for each task. For each of the ten trials, we varied the random seed which controlled the tasks (whenever the tasks were not fixed by definition), the random splits for training/validation/test, and the order in which the tasks were presented to the agent. Validation sets were only used by dynamic + compositional learners for selecting whether to keep a new component. Details are summarized in Table E.2.

## E.2 NETWORK ARCHITECTURES

We used $k = 4$ components for all compositional algorithms with fixed $k$. This is the only architectural choice for linear models. Below, we describe the architectures used for other experiments.

Table E.2: Data set details summary.

| | FERA | Landmine | Schools | MNIST | Fashion | CUB | CIFAR | Omniglot |
|---|---|---|---|---|---|---|---|---|
| tasks | 21 | 29 | 139 | 10 | 10 | 20 | 20 | 50 |
| classes | 2 | 2 | — | 2 | 2 | 10 | 5 | 14–55 |
| features | 100 | 9 | 27 | 784 | 784 | 512 | $32\times32\times3$ | $105\times105$ |
| feat. extract. | PCA | — | — | — | — | ResNet-18 | — | — |
| train | 225–499 | 222–345 | 11–125 | $\sim$9500 | $\sim$9500 | $\sim$120 | $\sim$2000 | 224–880 |
| val | — | — | — | $\sim$2500 | $\sim$2500 | $\sim$30 | $\sim$500 | 28–110 |
| test | 225–500 | 223–345 | 11–126 | $\sim$2000 | 2000 | $\sim$150 | 500 | 28–110 |

**Soft layer ordering** We based our soft layer ordering architectures on those used by Meyerson & Miikkulainen (2018), whenever possible. For MNIST and Fashion, we used a random and fixed linear input transformation $\mathcal{E}^{(t)}$ for each task, and each component was a fully connected layer of 64 units. For CUB, all tasks shared a fixed ResNet-18 pre-trained on ImageNet[2] as an input transformation, followed by a task-specific input transformation $\mathcal{E}^{(t)}$ given by a linear trained layer, and each component was a fully connected layer of 256 units. For CIFAR, there was no input transformation, and each component was a convolutional layer of 50 channels with $3 \times 3$ kernels and padding of 1 pixel, followed by a max-pooling layer of size $2 \times 2$. Finally, for Omniglot, there was also no input transformation, and each component was a convolutional layer of 53 channels with $3 \times 3$ kernels and no padding, followed by max-pooling of $2 \times 2$ patches. The input images to the convolutional nets in CIFAR and Omniglot were padded with all-zero channels in order to match the number of channels required by all component layers (50 and 53, respectively). All component layers were followed by ReLU activation and a dropout layer with dropout probability $p = 0.5$. The output of each network was a linear task-specific output transformation $\mathcal{D}^{(t)}$ trained individually on each task. The architectures for jointly trained baselines were identical to these, and those for no-components baselines had the same layers but no mechanism to select the order of the layers.

**Soft gating** The soft gating architectures mimicked those of the soft layer ordering architectures closely, all having the same input and output transformations, as well as the same components. The only difference was in the structure architectures. For fully connected nets, at each depth, the structure function $s^{(t)}$ was a linear layer that took as input the previous depth's output and whose output was a soft selection over the component layers for the current depth. For convolutional nets, there was one gating net per task with the same architecture as the main network. The structure $s^{(t)}$ was computed by passing the previous depth's output in the main network through the remaining depths in the gating network (e.g., the output of depth 2 in the original network was passed through depths 3 and 4 in the gating network to compute the structure over modules at depth 3).

### E.3 ALGORITHM DETAILS

All agents trained for 100 epochs on each task, with a mini-batch of 32 samples. Compositional agents used the first 99 epochs solely for assimilation and the last epoch for adaptation. Dynamic + compositional agents followed this same process, but every assimilation step was done via component dropout; after the adaptation step, the agent kept the new component if its validation performance with the added component represented at least a 5% relative improvement over the performance without the additional component. Joint agents trained all components and the structure for the current task jointly during all 100 epochs, keeping the structure for the previous tasks fixed, while no-components agents trained the whole model at every epoch.

ER-based algorithms used a replay buffer of a single mini-batch per task. Similarly, EWC-based algorithms used a single mini-batch to compute the approximate Fisher information matrix required for regularization, and used a fixed regularization parameter $\lambda = 10^{-3}$.

To ensure a fair comparison, all algorithms, including our baselines, used the same initialization procedure by training the first $T_{\texttt{init}} = 4$ tasks jointly, in order to encourage the network to generalize across tasks. For soft ordering nets, the order of modules for the initial tasks was initialized as a random one-hot vector for each task at each depth, ensuring that each component was selected at

---

[2]The pre-trained ResNet-18 is provided by PyTorch, and we followed the pre-processing recommended at `https://pytorch.org/docs/stable/torchvision/models.html`.

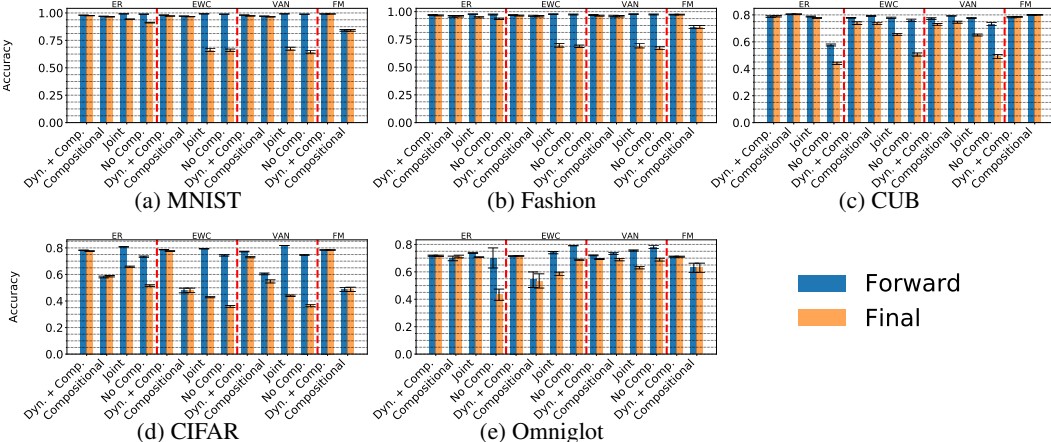

Figure F.2: Soft layer ordering accuracy. Compositional agents outperformed baselines in most data sets for every adaptation method. Dyn. + comp. agents further improved performance, leading to our methods being strongest. Error bars denote standard errors.

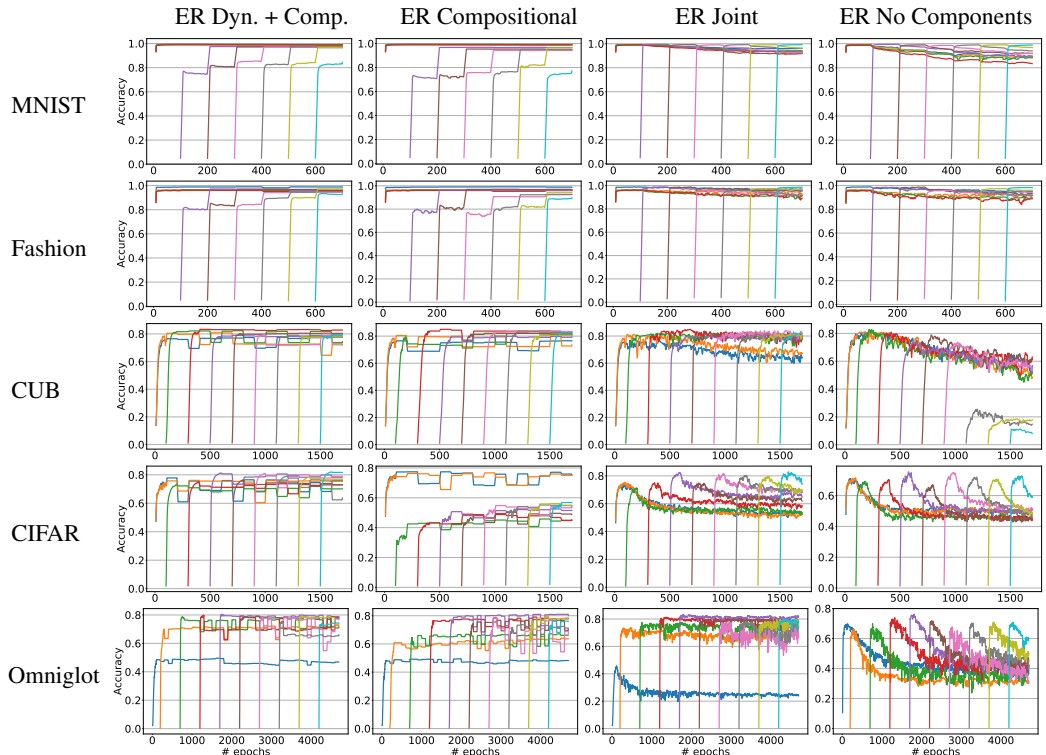

Figure F.3: Smoothed learning curves with soft layer ordering using ER. Compositional methods did not exhibit decaying performance of early tasks, while joint and no-components baselines did.

least once, and for soft gating nets, the gating nets were randomly initialized. The structures over initial tasks were kept fixed during training, modifying only the weights of the components.

## F  ADDITIONAL RESULTS FOR QUANTITATIVE EXPERIMENTS

We now present detailed results that expand upon those presented in Section 5.2 in the main paper.

For completeness, we include expanded results from Figures 1 and 2 in the main paper, corresponding to soft layer ordering. Figure F.2 is a more detailed version of Figure 1, and shows the test accuracy immediately after each task was trained and after all tasks had been trained, separately for each data set. Compositional algorithms conforming to our proposed framework achieve a better

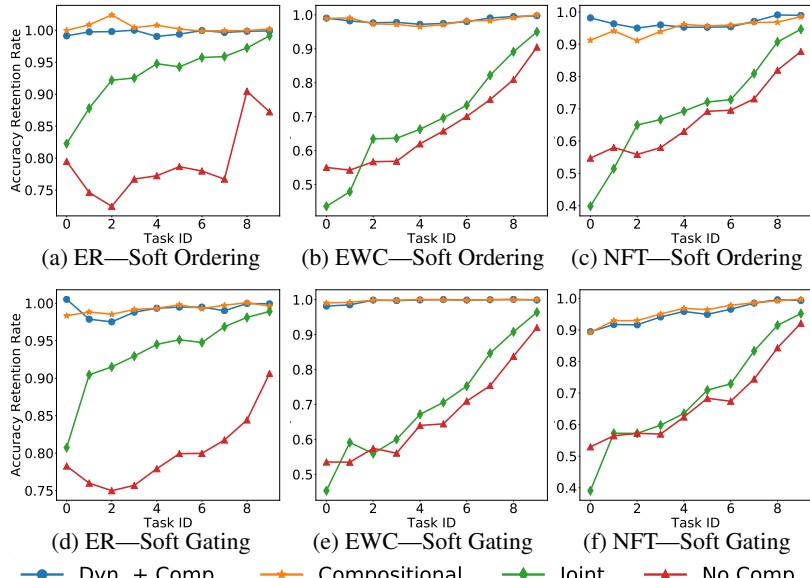

Figure F.4: Catastrophic forgetting across data sets. Ratio of accuracy when a task was first trained to when all tasks had been trained. For data sets with more than ten tasks, we sampled ten interleaved tasks to match all the x-axes. Compositional algorithms had practically no forgetting, whereas jointly trained and no-components baselines forgot knowledge required to solve earlier tasks.

Table F.3: Number of learned components. Standard errors reported after the $\pm$.

| Structure | Base | MNIST | Fashion | CUB | CIFAR | Omniglot |
|---|---|---|---|---|---|---|
| Soft ordering | ER | $5.2 \pm 0.3$ | $4.9 \pm 0.3$ | $5.9 \pm 0.3$ | $19.1 \pm 0.3$ | $9.3 \pm 0.3$ |
| | EWC | $5.0 \pm 0.3$ | $4.7 \pm 0.2$ | $5.8 \pm 0.2$ | $19.6 \pm 0.2$ | $10.1 \pm 0.3$ |
| | NFT | $5.0 \pm 0.2$ | $4.8 \pm 0.3$ | $6.1 \pm 0.3$ | $17.7 \pm 0.3$ | $10.0 \pm 0.7$ |
| | FM | $10.0 \pm 0.0$ | $8.8 \pm 0.2$ | $6.5 \pm 0.4$ | $19.1 \pm 0.4$ | $10.2 \pm 0.6$ |
| Soft gating | ER | $4.0 \pm 0.0$ | $4.2 \pm 0.1$ | — | $4.1 \pm 0.1$ | $7.1 \pm 0.4$ |
| | EWC | $4.1 \pm 0.1$ | $4.0 \pm 0.0$ | — | $4.8 \pm 0.2$ | $7.4 \pm 0.4$ |
| | NFT | $4.1 \pm 0.1$ | $4.2 \pm 0.1$ | — | $4.1 \pm 0.1$ | $7.2 \pm 0.3$ |
| | FM | $5.4 \pm 0.2$ | $4.7 \pm 0.2$ | — | $4.4 \pm 0.2$ | $7.3 \pm 0.4$ |

trade-off than others in flexibility and stability, leading to good adaptability to each task with little forgetting of previous tasks. Similarly, Figure F.3 shows learning curves similar to those in Figure 2 in the main paper, for each data set. Baselines that train components and structures jointly all exhibit a decay in the performance of earlier tasks as learning of future tasks progresses, whereas methods conforming to our framework do not. Results for soft gating nets display a similar behavior.

The gap between the first and second bar for each algorithm in Figure F.2 is an indicator of the amount of catastrophic forgetting. However, it hides details of how forgetting affects each individual task. On the other hand, the decay rate of each task in Figure F.3 shows how each task is forgotten over time, but does not measure quantitatively how much forgetting occurred. Based on prior work (Lee et al., 2019), we evaluated the ratio of performance after each task was trained to after all tasks had been trained as a metric of knowledge retention. Results in Figure F.4 show that compositional methods exhibit substantially less catastrophic forgetting, particularly for the earlier tasks seen during training.

In our experiments, it was in many cases necessary to incorporate an expansion step in order for our algorithm to be sufficiently flexible to handle the stream of incoming tasks. This expansion step enables our methods to dynamically add new components if the existing ones are insufficient to achieve good performance in the new task. Table F.3 shows the number of components learned by each dynamic algorithm using both soft ordering and soft gating, averaged across all ten trials. Notably, in the soft ordering case, in order for our methods to work on the CIFAR data set, they required learning almost one component per task. This explains why compositional algorithms

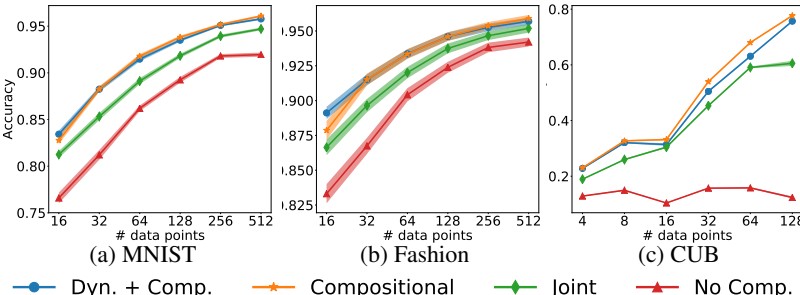

Figure F.5: Accuracy of ER-based methods with varying data sizes. Compositional methods performed better even with extremely little data per task. Shaded area represents standard errors.

Table F.4: Number of tasks that reuse a component. A task reuses a component if its accuracy drops by more than $5\%$ relative when the component is dropped. Standard errors reported after the $\pm$.

| Algorithm | Comp. | MNIST | Fashion | CUB | CIFAR | Omniglot |
|---|---|---|---|---|---|---|
| Compositional | 0 | $6.40 \pm 0.43$ | $6.00 \pm 0.24$ | $13.40 \pm 0.78$ | $18.90 \pm 0.30$ | $46.40 \pm 0.98$ |
| | 1 | $4.90 \pm 0.26$ | $4.70 \pm 0.28$ | $7.90 \pm 0.41$ | $16.20 \pm 0.80$ | $30.60 \pm 2.86$ |
| | 2 | $4.10 \pm 0.26$ | $4.10 \pm 0.17$ | $5.90 \pm 0.57$ | $11.90 \pm 1.14$ | $18.80 \pm 3.76$ |
| | 3 | $3.00 \pm 0.32$ | $2.60 \pm 0.32$ | $3.20 \pm 0.49$ | $5.70 \pm 0.97$ | $10.90 \pm 2.17$ |
| Dyn. + Comp. | 0 | $4.70 \pm 0.38$ | $5.10 \pm 0.26$ | $9.80 \pm 0.98$ | $13.30 \pm 1.27$ | $21.90 \pm 1.82$ |
| | 1 | $3.60 \pm 0.25$ | $3.90 \pm 0.22$ | $6.20 \pm 0.56$ | $6.20 \pm 0.61$ | $12.30 \pm 0.68$ |
| | 2 | $2.80 \pm 0.28$ | $3.30 \pm 0.28$ | $4.40 \pm 0.62$ | $4.00 \pm 0.35$ | $9.20 \pm 0.61$ |
| | 3 | $1.90 \pm 0.26$ | $2.10 \pm 0.22$ | $2.70 \pm 0.20$ | $3.10 \pm 0.26$ | $7.40 \pm 0.47$ |
| | 4 | — | — | — | $3.00 \pm 0.32$ | $6.50 \pm 0.49$ |
| | 5 | — | — | — | $1.80 \pm 0.13$ | $5.00 \pm 0.51$ |
| | 6 | — | — | — | $1.50 \pm 0.16$ | $4.20 \pm 0.46$ |
| | 7 | — | — | — | $1.10 \pm 0.09$ | $3.40 \pm 0.47$ |
| | 8 | — | — | — | $1.00 \pm 0.00$ | $1.90 \pm 0.30$ |
| | 9 | — | — | — | $1.00 \pm 0.00$ | — |
| | 10 | — | — | — | $1.00 \pm 0.00$ | — |
| | 11 | — | — | — | $1.00 \pm 0.00$ | — |
| | 12 | — | — | — | $1.00 \pm 0.00$ | — |
| | 13 | — | — | — | $0.90 \pm 0.09$ | — |
| | 14 | — | — | — | $0.90 \pm 0.09$ | — |

without dynamic component additions were incapable of performing well on CIFAR. It is also worth noting that soft gating nets typically required adding fewer new components, which is to be expected, since the gating structure gives the learner substantially more flexibility. Recall that, as mentioned in Section 5.2.3, soft gating networks were unable to perform well on the CUB data set because of the small sample size, so the corresponding column is omitted from the table.

One of the key aspects of lifelong learning is the ability to learn in the presence of little data for each task, using knowledge acquired from previous tasks to acquire better generalization for new tasks. To evaluate the sample efficiency of our algorithm, we varied the number of data points used for training for MNIST, Fashion, and CUB using the soft ordering structure and ER. We repeated the evaluation for 50 trials, each with a different random seed controlling the selection of classes and samples for each task, and the order over tasks. Learners were trained for 1,000 epochs, with our compositional methods alternating nine epochs of assimilation and one epoch of adaptation. We used a batch size of $b = 32$, and limited the replay buffer size to $\min(\max(\lfloor 0.1n \rfloor, 1), b)$ for each data size $n$. Figure F.5 shows the learning accuracy for ER-based algorithms as a function of the number of training points, revealing that compositional algorithms work better than baselines even in the presence of very little data.

Our approach was designed to discover a set of components that are reusable across multiple tasks. To verify that this effectively occurs, we evaluated how many tasks reuse each component. Taking the models pre-trained via compositional and dynamic + compositional ER with soft layer ordering, we evaluated the accuracy of the model on each task if any individual component was removed

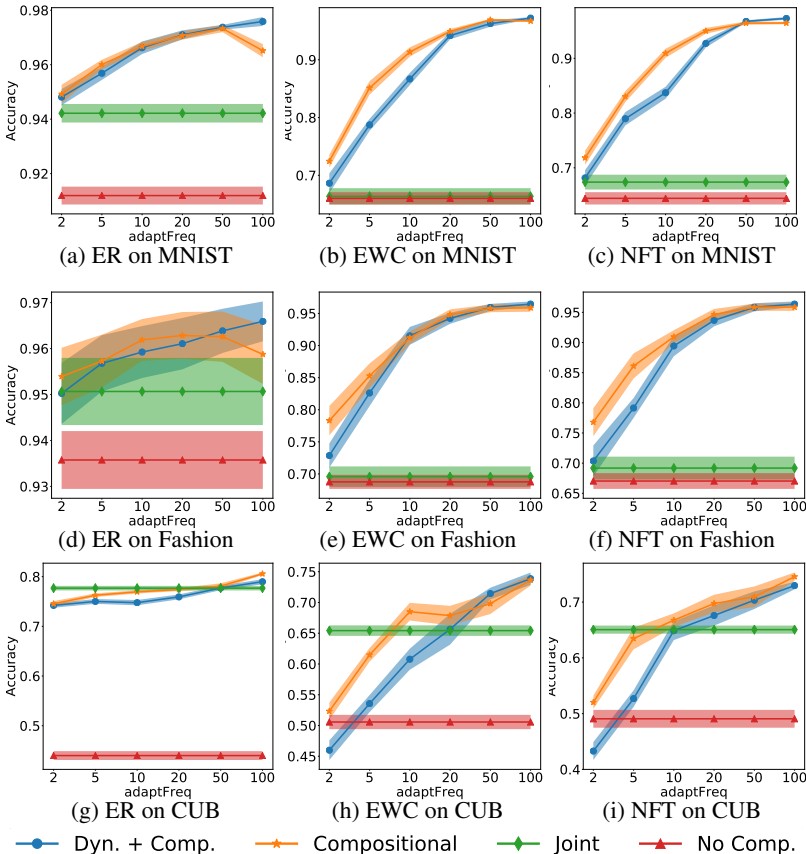

Figure F.6: Effect of the assimilation and accommodation schedule. Average accuracy across tasks w.r.t. the number of assimilation epochs between accommodation epochs. Broadly, methods under our framework performed better with a scheduled that favored stability, taking more assimilation steps before accommodating any new knowledge into the set of existing components.

from the model. We then considered a task to reuse a given component if removing it caused a relative drop in accuracy of more than $5\%$. Table F.4 shows the number of tasks that reuse each component. Since there is no fixed ordering over components across trials, we sorted each trial's components in descending order of the number of tasks that reused each component. Moreover, for dynamic + compositional ER, we only consider components that are created across all trials for a given data set to ensure all averages are statistically significant. We found that across all data sets and algorithms, all $k = 4$ components available from initialization were used by multiple tasks. For the Omniglot data set, we find that this behavior persists even for components that are dynamically added in the expansion step. However, this is not the case for the CIFAR data set, for which the first few dynamically added components are indeed reused by multiple tasks, but subsequent ones are used by a single task. This indicates that those components were added merely for increasing performance of that individual task, but found no reusable knowledge useful for future tasks.

When designing algorithms under our framework, one needs to choose how to alternate the processes of assimilation and accommodation. In most experiments so far, we considered the simplest case, where adaptation is entirely carried out after assimilation is finished. However, it is possible that other choices yield better results, enabling the learner to incorporate knowledge about the current task that further enables it to assimilate it better. To study this question, we carried out additional experiments using ER variants on the MNIST, Fashion, and CUB data sets with soft layer ordering. Instead of executing the adaptation step only after completing assimilation, we alternated epochs of assimilation with epochs of adaptation with various frequencies. Results are displayed in Figure F.6. Generally, we found that it was beneficial to carry out adaptation steps infrequently, with a clear increasing trend in performance as the learner took more assimilation steps before each adaptation step. For MNIST and Fashion, we found that all choices of schedule led to improved per-

Table G.5: Average final accuracy across tasks on the Combined data set. Each column shows accuracy on the subset of tasks from each given data set, as labeled. Standard errors after $\pm$.

| Base | Algorithm | All data sets | MNIST | Fashion | CUB |
|------|-----------|---------------|-------|---------|-----|
| ER | Dyn. + Comp. | $\mathbf{86.5 \pm 1.8}\%$ | $\mathbf{99.5 \pm 0.0}\%$ | $\mathbf{98.0 \pm 0.3}\%$ | $\mathbf{74.2 \pm 2.0}\%$ |
| | Compositional | $82.1 \pm 2.5\%$ | $\mathbf{99.5 \pm 0.0}\%$ | $97.8 \pm 0.3\%$ | $65.5 \pm 2.4\%$ |
| | Joint | $72.8 \pm 4.1\%$ | $98.9 \pm 0.3\%$ | $97.0 \pm 0.7\%$ | $47.6 \pm 6.2\%$ |
| | No Comp. | $47.4 \pm 4.5\%$ | $91.8 \pm 1.3\%$ | $83.5 \pm 2.5\%$ | $7.1 \pm 0.4\%$ |
| EWC | Dyn. + Comp. | $\mathbf{75.1 \pm 3.2}\%$ | $98.7 \pm 0.5\%$ | $\mathbf{97.1 \pm 0.7}\%$ | $\mathbf{52.4 \pm 2.9}\%$ |
| | Compositional | $71.3 \pm 4.0\%$ | $\mathbf{99.4 \pm 0.0}\%$ | $96.1 \pm 0.9\%$ | $44.8 \pm 3.5\%$ |
| | Joint | $52.2 \pm 5.0\%$ | $85.1 \pm 5.5\%$ | $88.6 \pm 3.8\%$ | $17.5 \pm 1.5\%$ |
| | No Comp. | $28.9 \pm 2.8\%$ | $52.9 \pm 1.6\%$ | $52.5 \pm 1.4\%$ | $5.0 \pm 0.4\%$ |
| NFT | Dyn. + Comp. | $\mathbf{75.5 \pm 3.2}\%$ | $\mathbf{99.1 \pm 0.3}\%$ | $\mathbf{96.2 \pm 0.9}\%$ | $\mathbf{53.3 \pm 2.8}\%$ |
| | Compositional | $70.6 \pm 3.8\%$ | $98.5 \pm 0.5\%$ | $95.6 \pm 0.8\%$ | $44.2 \pm 3.5\%$ |
| | Joint | $52.7 \pm 4.9\%$ | $85.5 \pm 4.9\%$ | $88.5 \pm 3.7\%$ | $18.4 \pm 1.7\%$ |
| | No Comp. | $34.6 \pm 3.7\%$ | $61.3 \pm 3.8\%$ | $59.8 \pm 3.6\%$ | $8.7 \pm 0.5\%$ |
| FM | Dyn. + Comp. | $\mathbf{83.8 \pm 2.0}\%$ | $\mathbf{99.6 \pm 0.0}\%$ | $\mathbf{98.3 \pm 0.3}\%$ | $\mathbf{68.7 \pm 1.5}\%$ |
| | Compositional | $74.6 \pm 3.1\%$ | $99.5 \pm 0.0\%$ | $98.1 \pm 0.3\%$ | $50.3 \pm 2.0\%$ |

formance over baselines, highlighting the benefits of splitting the learning process into assimilation and accommodation. For CUB, the results were more nuanced, with very fast accommodation rates achieving lower accuracy than the baselines. This is consistent with our results in Table 3, where compositional FM, equivalent to compositional ER with a schedule of infinite assimilation steps per accommodation step, performed nearly as well as compositional ER with a single adaptation epoch.

## G  COMPLETE RESULTS ON SEQUENCES OF DIVERSE TASKS

We now describe in more detail the experimental setting used to obtain the results in Section 5.2.4 in the main paper, and provide the complete table of results using all our instantiations and baselines on the Combined data set.

We combined the 10 MNIST tasks, 10 Fashion tasks, and 20 CUB tasks into a single lifelong learning data set with $T = 40$ tasks, and trained all our methods on this new Combined data set. None of the methods were informed in any way that the tasks came from different data sets, and each learner was simply required to learn all tasks consecutively exactly as in the remaining experiments. We used the soft layer ordering structure, with the architecture used for the CUB tasks described in Appendix E. Only CUB images were processed with the pre-trained ResNet-18, whereas MNIST and Fashion images where fed directly to the task-specific input transformation $\mathcal{E}^{(t)}$.

The results are summarized in Table G.5. As expected, our methods clearly outperformed all baselines, by a much wider margin than in the single-data-set settings of Section 5.2.2 in the main paper. In particular, no-components baselines (those with monolithic architectures) were completely incapable of learning to solve the CUB tasks. Even the jointly trained variants, which do have compositional structures but learn them naïvely with existing lifelong methods, failed drastically. Our methods were far better, especially when using ER as the base adaptation method.

Note that, in order to match the requirements of the CUB data set, the architecture we used gave MNIST and Fashion a higher capacity (layers of size 256 vs 64) and the ability to train the input transformation for each task individually (instead of keeping it fixed) compared to the architecture described in Appendix E. This explains the higher performance of most methods in those two data sets compared to the results in Table 3 in the main paper.

## H  VISUALIZATION OF THE LEARNED COMPONENTS

The primary motivation for our framework was the creation of lifelong learning algorithms capable of discovering self-contained, reusable components, useful for solving a variety of tasks. In this section, provide additional details about the visualization experiment of Section 5.3, as well as a more comprehensive study of various components and a comparison to the joint ER baseline.

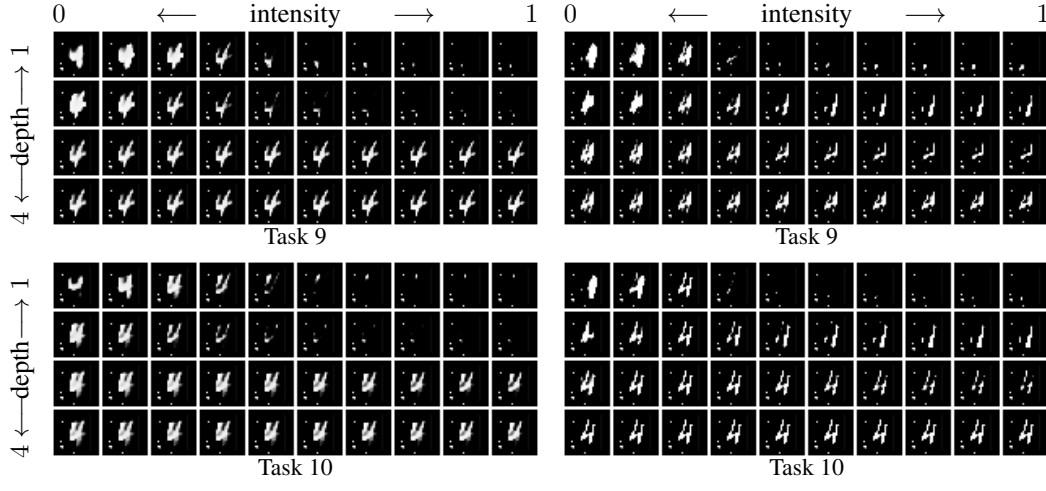

(a) ER Compositional                    (b) ER Joint

Figure H.7: Visualization of reconstructed MNIST "4" digits on the last two tasks seen by the compositional and joint variants of ER with soft layer ordering, varying the intensity with which component $i = 0$ is selected. Compositional ER learned a component that performs a functional primitive: the more intensely the component is selected (moving from left to right on each row), the thinner the lines of the digit become. The magnitude of this effect decreases with depth (moving from top to bottom), with the digit completely disappearing as the component is more intensely selected at the earliest layers, but only becoming slightly sharper with intensity at the deepest layers. This effect is consistent across both tasks. Joint ER did not exhibit this consistent behavior, with different effects observed at different depths and for the different tasks.

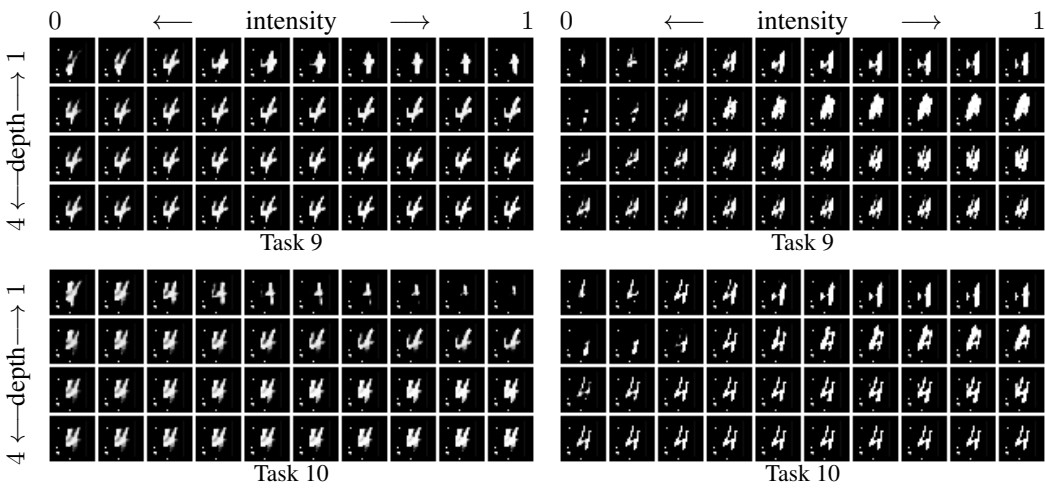

(a) ER Compositional                    (b) ER Joint

Figure H.8: Visualization of reconstructed MNIST "4" digits, varying the intensity of component $i = 1$. The component learned via compositional ER consistently decreases the length of the left side of the digit and increases that of the right side. Again, we were unable to detect any consistency in the effect of the component learned via joint ER.

We followed the visualization experiment of Meyerson & Miikkulainen (2018), where each task corresponded to a single image of the digit "4", and each pixel in the image constituted one datapoint. The $x, y$ coordinates of the pixel were used as features, and the pixel's intensity was the associated label. Pixel coordinates and intensities were normalized to $[0, 1]$. All pixels in the image were treated as training data, since we were interested in understanding the learned representations, as opposed to generalizing to unseen data. Our network had $k = 4$ components shared across all tasks, and used soft layer ordering to learn the structure $s^{(t)}$ for each task. We used a linear input transformation layer $\mathcal{E}^{(t)}$ shared across all tasks, and a shared sigmoid output transformation layer

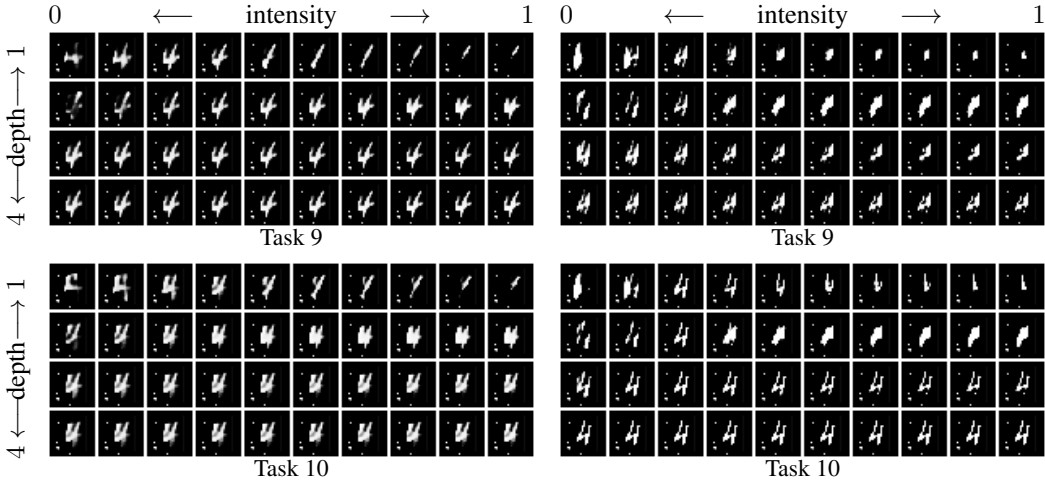

(a) ER Compositional          (b) ER Joint

Figure H.9: Visualization of reconstructed MNIST "4" digits, varying the intensity of component $i = 2$. As the intensity of the component learned via compositional ER increased, the digit changed from very sharp to very smooth. Joint ER again did not exhibit any consistent behavior.

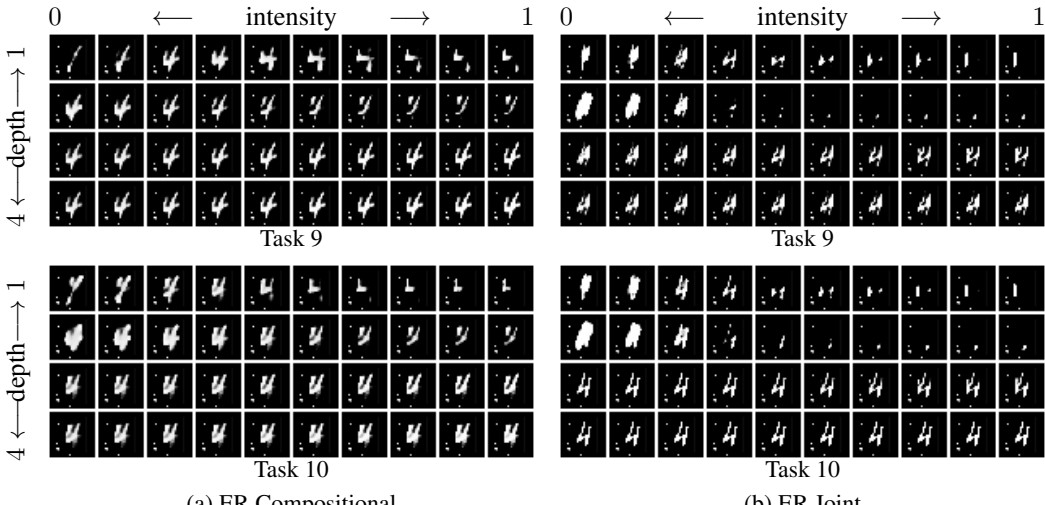

(a) ER Compositional          (b) ER Joint

Figure H.10: Visualization of reconstructed MNIST "4" digits, varying the intensity of component $i = 3$. This component also interpolates between sharper and smoother digits, while also rotating the digit. There was no consistency in the behavior of the component learned by Joint ER.

$\mathcal{D}^{(t)}$. Sharing the input and output transformations across tasks ensures that the only differences across the models of the different tasks are due to the structure of each task over the components. We trained the network to minimize the binary cross-entropy loss on $T = 10$ tasks for 1,000 epochs via the compositional and jointly trained versions of ER with a replay buffer and batch size of 32 pixel instances, updating the components of the compositional version every 100 epochs.

To evaluate the ability of compositional ER to capture reusable functional primitives, we observed the reconstructed images output by our network as we varied the intensity $\psi_{i,j}^{(t)}$ with which one specific component $i$ is chosen at different depths $j$ in the network. We focused our evaluation on the last two tasks seen by the learner, in order to disregard the effects of catastrophic forgetting, which rendered the visualizations of the outputs of the joint ER baseline incomprehensible for earlier tasks. Figures H.7–H.10 show these reconstructions as the intensity of each component individually varies at different depths. The components learned with compositional ER learned to produce effects on the digits consistent across tasks, with more extreme effects at the initial layers. In contrast, joint ER learned components whose effects are different for different tasks and at different depths.

