# OpenReview forum: "Lifelong Learning of Compositional Structures"
_ICLR.cc/2021/Conference — ICLR 2021 Poster_

### Official Review · AnonReviewer4 · 2020-10-25
**Excellent paper on lifelong learning**

**Rating:** 9
**Confidence:** 4

**Review:**

Summary:

The paper introduces a framework for lifelong learning of compositional structures. The algorithm is loosely inspired by biological learning and consists of two main steps. The step of component selection relies on existing methods that can learn task-specific structure. In the next step (adaptation), the algorithm adapts the knowledge from the previous tasks to the current task and if that is insufficient to solve the task, new components are added. Adaptation step relies on existing methods for adapting the knowledge state given a new task in continual learning (component parameters are updated). Knowledge expansion (adding new components) uses component dropout, a method proposed by the authors which combines pruning and alternating backpropagation steps with and without the potential new component. The proposed method is beneficial in terms of computational complexity in comparison with the standard lifelong learning methods. The authors evaluate the method on three compositional structures and show that it outperforms the baselines. The paper includes visualisation of the learned components, extensive appendix with additional experiments and ablation studies, and a systematic overview of the prior work in learning compositional structures and lifelong learning.

Score justification:

The paper is exceptionally well-written and rich in terms of experimental results and ablation studies. The proposed algorithm combines existing methods in a novel way and extends them with component dropout. The topics of learning compositional structures and lifelong/continual learning are of high interest to the community. The new procedure (Algorithm 1) balances out the key problems of retaining existing knowledge and expanding it without frequent expensive data revisiting steps. The only concern would be the lack of conclusion and sporadic vague phrasing. Overall the paper is very interesting to read and it should have a strong impact on the research in lifelong learning.

Pros:

The justified parallel between the algorithm and children development research

The contributions are well-positioned in comparison to the existing work (in the Related work section and throughout the paper)

Computational complexity is discussed

Extensive experiments which show strong results in learning components of an increasing complexity

Code is released for reproducibility

Cons:

The main paper combined with the appendix is quite long for a conference paper

Conclusion is missing

From the sentence “Our framework separates the learning process into two broad stages:learning how to best combine existing components in order to assimilate a novel problem, and learning how to adapt the set of existing components to accommodate the new problem.” it is unclear how the two stages differ

In Table 1, you could mention the meaning of the symbols which parametrize the complexities in the caption so that it is possible to better understand the computational complexity analysis without the appendix

Questions during rebuttal period:

Please address and clarify the cons above

Typos: None were found

---

> ### Author Response · Authors · 2020-11-14
> **Thank you. We have addressed your comments below.**
>
> Thank you for you very high assessment of our work. We appreciate the feedback, and have incorporated your suggestions into our final draft. Concretely, we added a conclusion section, which we had omitted due to space restrictions (page 9). We also added the notational symbols to the caption of Table 1 (page 5, middle of page) to improve readability. We have highlighted these and other changes by using blue font.
>
> If you have outstanding questions during the discussion period, we'd be happy to address them.

---

### Official Review · AnonReviewer1 · 2020-10-26
**Review for submission**

**Rating:** 6
**Confidence:** 3

**Review:**

This paper addresses lifelong learning of compositional structures by proposing a general-purpose framework, which separates the learning process to two stages: combine existing components for assimilation; adapt existing components for accommodation and optionally add new components.

The strong point is that the approach is general for model architectures and can be combined with different catastrophic forgetting mechanisms. There are many evaluation datasets and settings. Also, it has visualization to analyze the learnt models. The weak point is that the arguments and evaluations do not support the claims very well.

My recommendation is that this paper is below the acceptance bar.

The reasons are mainly from concerns for compositional structure learning and catastrophic forgetting evaluation.
1. The paper does not clearly explain why separating assimilation and accommodation leads to learning compositional structure and components from a machine learning perspective. Even an intuitive explanation or a concrete example may be more helpful.
2. The evaluation does not directly support that compositional the structure is learnt. The main evaluation metric is the final accuracy, which may tell that the proposed approach works as a good pre-training algorithm for reusable modules, but this does not mean the compositional structure and components are learnt. Though there are visualizations, it still lacks numerical evaluation for this purpose.
3. If this paper emphasizes ability as a pre-training algorithm, it should compare with other pre-training algorithms.
4. This paper does not explain why the proposed approach is helpful for catastrophic forgetting.
The experiment setting has the first 99 epochs solely for assimilation and the last epoch for adaptation. So a possible explanation is that the proposed approach has 99% less iteration to update the parameters then baselines, hence it better avoids catastrophic forgetting, and just stores task specific parameters. This seems trivial and is not a stand-alone significant contribution.

---

> ### Author Response · Authors · 2020-11-14
> **Thank you for your feedback. We address the main points in your review below. (1/2)**
>
> 1. Splitting the learning process into assimilation and accommodation stages is useful specifically in the lifelong setting. Intuitively, you want to maximize the reuse of existing knowledge as much as possible, and then adapt the structural representation to accommodate new knowledge. This intuition has been leveraged in past lifelong learning work (e.g., Ruvolo and Eaton, 2013), but it was never tested whether this split is useful. Our ablative experiments clearly show that it is.
>
>   In a multi-task setting, where all tasks are available for simultaneous training, the learner can simply modify components and structures simultaneously until it converges to a reasonable decomposition of knowledge the works for all tasks. However, in a lifelong setting, the problem is how to decompose this knowledge without ever seeing multiple tasks jointly. If the learner attempts to adapt the components to the new task simultaneously as it attempts to discover how they can be combined to solve the new task, it might catastrophically damage the components in the process (essentially by overfitting them to the current task). We observe this behavior in our experiments, even if using techniques for avoiding forgetting like EWC or ER.
>
>   As an illustrative example, imagine a scenario where the tasks involve detecting circles or squares in varying background noise conditions. Intuitively, we would want our main components to be a circle detector and a square detector. However, if the first two tasks are "circle without noise" and "square with noise", the components would likely be initialized to overfit to those combinations. So, at this stage we have a "circle + no-noise" component and a "square + noise" component. Let's say the agent is now faced with the task "circle with noise".
>   - It is entirely possible that the learner initially considers the "square + noise" detector to be more suitable for the current task (e.g., due to random initialization), and therefore modifies it slightly to be more useful for the current task. Since the module will get better at detecting the circle, it will continue modifying this component until it becomes a "(square or circle) + noise" detector. This is clearly not what we want.
>   - With assimilation / accommodation, even if the learner initially selects the "square + noise" detector, gradient descent will likely eventually discover that the "circle + no-noise" is better suited for the current task, even if it's not perfect for it. Critically, this is only possible because the learner _has not_ modified the existing components to better suit the current task. Then, once the learner knows which components are relevant for the current task, it can adapt them to better generalize. This would turn the "circle + no-noise" detector into the desired "circle" detector.
>
> 2. We are not sure we have properly understood what you mean by our method being only a good pre-training approach. If you mean that the components learned during the initialization stage (step 1 of our framework) are good enough to solve future tasks, we ablate this by comparing against FM compositional, which re-uses these initialized components but does not adapt them in any way to the new task. Moreover, joint training variants also use this initialization step, but despite a good set of initial components being available, joint methods still are not able to solve the problem well. In terms of measuring compositionality explicitly, creating a metric that measures how well knowledge is decomposed is an open problem, and we would certainly be interested in studying how our method performs under such a metric. As proxies, we measure:
>   - Visualizations in Figure 3 and Appendix G. As you point out, these do not explicitly measure _how much_ composition there is, but they suggest that the components are achieving task-independent and depth-independent knowledge, which is precisely what we hoped. These same experiments were carried out by Meyerson and Miikkulainen (2018) to show that soft layer ordering nets exhibited this behavior in the batch multi-task learning setting.
>   - Performance on an explicitly compositional data set in Appendix D. These results show that our methods are better than all baselines if considering this particular case where our compositional assumption holds explicitly.
>   - The number of tasks that use each component in Table F.4. This table confirms that all components are reusable across multiple tasks. In our response to R2, we showed that this behavior is also found in our experiments in Appendix D on a compositional data set.
>
>
> 3. We reiterate that our method is FAR more than a pre-training method, and as such we do not believe a comparison to pre-training techniques would be relevant.
>
> (continues below)

---

> > ### Author Response · Authors · 2020-11-14
> > **Thank you for your feedback. We address the main points in your review below. (2/2)**
> >
> > 4. You are correct that using a single epoch for modifying the shared components is a big reason why our methods avoid forgetting. This is precisely what we advocate for: splitting the learning process into assimilation and accommodation enables the learner to discover knowledge about the current task _before_ making modifications to the components that could affect the earlier tasks.
> >
> >   Of course, all baselines could also avoid forgetting if they used a single epoch of adaptation. However, they would certainly not be able to learn good models for most of the tasks with such little training. For completeness, we ran experiments to validate this trivial claim on all data sets, for both soft ordering and soft gating. Concretely, after initialization on four tasks for 100 epochs, all subsequent tasks are trained for a single epoch. Tables 1 and 2 below clearly show that a single epoch, even after the components are initialized, is insufficient for learning. We conclude that our method _does_ avoid forgetting due to using a single epoch of training, but this is only possible due to our assimilation/accommodation split, which enables acquiring knowledge _before_ modifying the existing components. This question of how to trade-off flexibility (to adapt to new tasks) and stability (to retain knowledge of previous tasks) is at the heart of lifelong learning research.
> >
> > Table 1: Average accuracy across tasks, training joint and no-components variants of soft layer ordering for a single epoch.
> >
> > | Base  | Algorithm    | MNIST        | Fashion     | CUB           | CIFAR        | Omniglot   |
> >
> > |:--------|:-----------------|:---------------|:--------------|:---------------|:--------------|:---------------|
> >
> > | ER      | Joint               | 94.8±0.4% | 94.3±0.8% | 26.2±0.4% | 36.9±1.1% | 13.1±2.1%  |
> >
> > | ER      | No Comp.     | 93.0±0.5% | 93.3±0.8% | 33.5±0.5% | 55.5±1.0% | 40.4±0.8%  |
> >
> > | EWC  | Joint               | 94.5±0.4% | 94.3±0.9% | 25.2±0.3% | 31.2±1.6% | 6.1±0.4%    |
> >
> > | EWC  | No Comp.     | 92.7±0.5% | 92.3±1.0% | 23.0±0.3% | 53.8±1.6% | 38.9±0.4%  |
> >
> > | VAN   | Joint               | 94.4±0.5% | 93.4±0.9% | 24.2±0.5% | 38.1±1.3% | 9.7±0.5%    |
> >
> > | VAN   | No Comp.     | 92.3±0.4% | 91.1±1.6% | 24.0±0.5% | 47.2±1.3% | 35.9±0.9%  |
> >
> > Table 2: Average accuracy across tasks, training joint and no-components variants of soft gating for a single epoch.
> >
> > | Base  | Algorithm    | MNIST       | Fashion      | CIFAR        | Omniglot   |
> >
> > |:--------|:-----------------|:--------------|:---------------|:---------------|:---------------|
> >
> > | ER      | Joint               | 91.3±0.5% | 90.7±1.5% | 60.2±0.5% | 33.2±1.1%  |
> >
> > | ER      | No Comp.     | 93.0±0.5% | 93.3±0.8% | 55.5±1.0% | 40.4±0.8%  |
> >
> > | EWC   | Joint              | 90.6±0.7% | 89.9±1.6% | 59.7±0.7% | 23.4±0.6%  |
> >
> > | EWC   | No Comp.    | 92.7±0.5% | 92.3±1.0% | 53.8±1.6% | 38.9±0.4%  |
> >
> > | VAN   | Joint               | 90.6±0.7% | 90.5±1.7% | 50.0±0.8% | 18.2±0.6%  |
> >
> > | VAN   | No Comp.     | 92.3±0.4% | 91.1±1.6% | 47.2±1.3% | 35.9±0.9%  |

---

> > > ### Comment · AnonReviewer1 · 2020-11-23
> > > **Thank you for reply**
> > >
> > > Thank you for answering the questions and updating the results with more experiments. I think they are clear and make sense, so I updated the score.

---

### Official Review · AnonReviewer3 · 2020-10-28
**Thorough contribution, which, maybe, lacks a "cherry on top".**

**Rating:** 7
**Confidence:** 3

**Review:**

** Summary **

The paper proposes a novel approach to lifelong learning, which builds upon the idea of gradually construcing a set of reusable components.
Through extensive experiments on different underlying architectures, the authors demonstrate the promise of their approach.

** Strengths **

The paper addresses a highly relevant problem.
The paper provides a very thorough evaluation of the proposed approach in a range of conditions.
The paper is very well written and is a pleasure to read.
In general, the contribution approaches the ideal when it comes to result reporting, ensuring reproducibility, and the thoroughness of experimental evaluation.

** Weaknesses **

While extremely thorough, the experiments are somewhat repetitive. I.e. overall we can confidently say that the method performs well on the selected tasks, but it seems that these experiments do not fully investigate the potential of the proposed architecture in diverse enough conditions.

The main selling point of the contribution is the ability to acquire compositional problem-solving components, but the qualitative analysis into whether the problem-solving components are actually compositional is very limited. The results given in appendix G and on the figure 3 are not very convincing, in my opinion. Similarly, in appendix D, when the architecture is applied in an intentionally compositional setting, the authors only report performance metrics.

Overall, it remains largely unclear whether the proposed method is truly leveraging compositionality, or is more of an efficient way of constructing an ensemble of classifiers.

Nevertheless, I believe that this paper remains above the acceptance threshold, despite these limitations.

** Typos **

"Qualitatively, we show that the components learned by an algorithm that subscribes our framework correspond to self-contained, reusable functions." - "subscribes" seems like a typo in this context.

** Questions **

- Intuitively, it seems that using component dropout introduces a conflict with the goal of obtaining disentangled/compositional components.
I.e. the authors say "Intermittently bypassing the new component ensures that existing components can compensate for it if it is discarded",
but at the same time it seems that for compositionality, we want different components to be unique and independent of each other. I would like the authors to clarify their reasoning behind using component droupout.

** Conclusion **

The paper is an extremely thorough and a very polished piece of work, which compensates what it somewhat lacks in "wow-effect" with general thoroughness and high quality of experiments, writing, and reporting.
After reading the paper I felt genuinely grateful to authors for putting this work together in such a thorough and thoughtful manner.

Overall, in my opinion, the contribution is above the acceptance threshold.

** Suggestions **

The paper abruptly ends with no summary or conclusion, which, in my opinion, severely affects its perception. While I understand that
this is due to the strict page limit, in my opinion, sacrificing conclusion is not reasonable. It is possible to move of the tables 3 or 4 to the appendix,
since they convey the same general message (the message being that the model performs well).

If the paper is not accepted to publication now, I would suggest moving some of the experimental evaluations tables into appendix and putting more effort into testing the model on qualitatively different tasks. It would be very interesting to see how the model performs on nontrivial problems requiring compositional reasoning,
and maybe less directly rooted in the visual domain. Even if staying in the visual domain is desirable, there is still room for improvement. For example, maybe would be possible to adapt the model to the RAVEN (https://arxiv.org/abs/1903.02741) dataset (after adapting it to the lifelong-learning setting) or similar tasks.

** Update after rebuttal **

I appreciate that the authors addressed my comments. After reading the authors response and other reviews, I still believe that it is a good paper that should be accepted to the conference.

---

> ### Author Response · Authors · 2020-11-14
> **Thank you for your high assessment of our work. We address your questions below.**
>
> - _Unclear whether approach leverages composition:_ While we agree that we don't directly measure whether our model explicitly exploits composition, this is in large part due to the lack of existing benchmarks and metrics related to composition, especially in the lifelong learning setting. However, we take steps towards understanding the compositional nature of our approach with: the visualizations in Figure 3 and Appendix G, the evaluations on the Objects data set in Appendix D (both of which you point out), and the measure of component reuse in Table F.4 in Appendix F and Table 1 in our response to R2. If you have any recommendation for metrics that could further the usefulness of the results in Appendix D, we'd be happy to report those, but we felt that replicating the analysis made for other benchmarks in this setting would make the paper far too repetitive, and the appendices even longer than they are already.
> - _Conflicting objective of component dropout:_ This is an excellent point. Intuitively, dropout does encourage the new component to be "replaceable" by the existing components. If it is possible to learn a model that can discard the current component without sacrificing performance, then dropout will require the learner to find it. If, on the other hand, the existing components are not sufficient, then dropout will fail to find a model where the component can be dropped, and so it will contain distinct knowledge from that present in existing components. Crucially, only the latest component is dropped out during training, and so existing components are encouraged to make up for this component, but not vice-versa. This means that the new component will not be encouraged to contain any knowledge already present in the existing ones.
> - We have added a conclusion to our revised draft.
> - Thanks for the reference to the RAVEN data set. We will look into it and explore it for future work.

---

### Official Review · AnonReviewer2 · 2020-10-28
**Good framework, would be interesting to have more results specific to the compositional property of the model.**

**Rating:** 6
**Confidence:** 4

**Review:**

The authors propose a new framework for compositional lifelong learning. In the proposed approach, the composition and adaptation parts are separated when a lifelong learner faces a new task: first, learn the best way to compose all existing components for the new task (and train an optional new component if exiting components aren't sufficient to reach a good performance), and only then adapt the components parameters to better fit the new problem. This new framework is validated on extensive experiments, using three composition and 3 adaptation strategies from the literature on 9 datasets. The paper is pleasing to read, each choice is discussed and justified

My main concern is about the scalability and the resilience to harder streams of tasks. Compositional and dynamic approaches are necessary when tackling complex streams of tasks, for example when having to deal with different modality or a large number of tasks.
All experiments in the main paper use relatively short sequences of tasks in which it makes sense to reuse all components on the new tasks since all tasks are strongly related (all coming from the same data set). I would be interested to see the performance of standard continual learning techniques from the literature on the same streams of tasks to demonstrate that there is indeed a gain in using this approach even on standard streams.
I would also be interested in seeing how sharp the component selection is on each task, is the assimilation step selecting only one or two components per layer or is it blending them all? We could expect a sharp component selection on the compositional dataset (appendix D), which would confirm that the different components are indeed specialized to handle specific properties of the tasks.

Minor questions and remarks:
- Appendix E.1 mentions that the validation set is only used for selecting whether to keep the new component. How are the hyper-parameters ( $\tau$, learning rates, number of samples to keep around for ER, $\lambda$, $T_{init}$, number of assimilation/adaptation epochs, ...) selected?
- Some papers on dynamic/compositional methods for continual learning should be discussed in the related work [1, 2, 3]


[1] Lifelong Learning with Dynamically Expandable Networks. Yoon et. al. [2017]

[2] Learn to Grow: A Continual Structure Learning Framework for Overcoming Catastrophic Forgetting. Li et. al. [2019]

[3] An Adaptive Random Path Selection Approach for Incremental Learning. Rajasegaran et. al. [2019]

Edit: Upgrade from 5 to 6 after the author's response.

---

> ### Author Response · Authors · 2020-11-14
> **Thanks for your feedback. We show that our methods are stronger than standard lifelong learners in standard streams.**
>
> - _Harder streams of tasks:_ We agree that the strength of compositional methods will be more apparent when evaluated on longer and harder streams of tasks. Unfortunately, to date there is no such benchmark that we are aware of, so we focus our evaluation on standard existing benchmarks and demonstrate that compositional learning achieves far superior performance even in this setting.
>
> - _Show that existing techniques are worse in this standard setting:_ As we pointed out in our response to R5, this is the purpose of no-components baselines. In particular, the no-components variants of EWC and ER, two popular continual learning methods, learn standard, monolithic architectures shared across tasks (with additional task-specific input and output mappings), and we show that they vastly underperform our methods. To substantiate these claims and compare against more existing non-compositional (or standard) methods, we included a comparison against CURL (Rao et al., 2019) in our response to R5, showing once more that compositional methods outperform them.
>
> - _Sharpness of component selection:_ We did not incorporate any procedure to encourage our selection functions to be sparse (e.g., Gumbel soft-max or sparsity regularization), so we don't necessarily expect the selection functions themselves to exhibit sparsity. However, Table F.4 in Appendix F shows the number of tasks that reuse each component. This number is clearly much lower than the total number of tasks, demonstrating that components are not all blended together for all tasks. The equivalent table for the Objects data set is included below, showing a similar trend.
>
> Table 1: Number of tasks that reuse a component on compositional data set. A task reuses a component if its accuracy drops by more than 5% relative when the component is dropped. Standard errors reported after the ±.
>
> |  Algorithm        | Component | Objects-Circle   | Objects-TopLeft   | Objects-Orange   | Objects-Random |
>
> |:---------------------|:----------------|:----------------------|:------------------------|:------------------------|:------------------------|
>
> | Compositional |   0                 |     5.40 ± 0.49      |      6.10 ± 0.50        |      6.00 ± 0.65       |       5.80 ± 0.88       |
>
> | Compositional |   1                 |     3.70 ± 0.28      |      4.60 ± 0.35        |      4.00 ± 0.24       |       3.30 ± 0.20       |
>
> | Compositional |   2                 |     3.00 ± 0.28      |      3.20 ± 0.28        |      3.10 ± 0.22       |       2.80 ± 0.24       |
>
> | Compositional |   3                 |     2.10 ± 0.33      |      2.00 ± 0.20        |      2.10 ± 0.22       |       1.90 ± 0.22       |
>
> | Dyn. + Comp.   |   0                 |     5.80 ± 0.85      |      6.30 ± 0.72        |      6.30 ± 0.93       |       5.70 ± 0.95       |
>
> | Dyn. + Comp.   |   1                 |     3.50 ± 0.21      |      3.90 ± 0.22        |      3.80 ± 0.24       |       3.30 ± 0.20       |
>
> | Dyn. + Comp.   |   2                 |     2.70 ± 0.20      |      2.80 ± 0.13        |      2.80 ± 0.13       |       3.10 ± 0.22       |
>
> | Dyn. + Comp.   |   3                 |     2.00 ± 0.28      |      2.00 ± 0.20        |      1.80 ± 0.19       |       2.00 ± 0.24       |
>
> - _Hyper-parameter choices:_ We did not spend substantial effort towards optimizing hyper-parameters. We considered the number of epochs, the size of the replay buffer, and the number of initialization tasks to be budgets, rather than tuneable hyper-parameters, and those were common to our methods and baselines. The only hyper-parameters of baselines are $\lambda$ for EWC, which we set to a simple default value (and checked whether increasing it or decreasing it improved performance of baselines---it didn't), and the learning rate (which we also checked worked well for baselines given the epoch budget). Our model-specific hyperparameters (\tau and assimilation/adaptation epochs) were chosen again as very simple values, and never varied during development. We studied the effect of the assimilation/adaptation schedule in Figure F.6 in Appendix F.
>
> - _Additional references:_ Thanks for pointing out those additional references. We have included the following comparison to those in Section 2: "Our framework also permits incorporating new components over time. Related work has increased network capacity in the non-compositional setting (Yoon et al., 2018) or in a compositional setting where previously learned parameters are kept fixed (Li et al., 2019). Another method enables adaptation of existing parameters, but requires expensively storing and training multiple models for each task to select the best one before adapting the existing parameters, and is designed for a specific choice of architecture, unlike our general framework (Rajasegaran et al., 2019)" (page 2, bottom of page)

---

> > ### Author Response · Authors · 2020-11-18
> > **We have run additional experiments, showing that our methods are far better than standard lifelong learners on harder streams of tasks.**
> >
> > We decided to run a simple test to verify that indeed compositional structures are most useful for harder streams of tasks, as you suggested (and as we expected). We combined the 10 MNIST, 10 Fashion, and 20 CUB tasks into a single 40-task lifelong learning data set, and trained all our methods on this new "Combined" data set. None of the methods were informed in any way that the tasks came from different data sets, and were simply required to learn all 40 tasks consecutively exactly as in our previous settings. We used the soft layer ordering structure, with the architecture used for the CUB tasks described in Appendix E.2.
> >
> > As we expected, our methods _highly_ outperformed all baselines, much more than in the "standard" settings. In particular, no-components baselines (those with monolithic or "standard" architectures) were completely incapable of learning to solve the CUB tasks. Even the jointly trained variants, which do have compositional structures but learn them naively with existing lifelong methods, failed drastically. Our methods were far better, especially when using ER as the base lifelong method. The results are summarized in the table below, and have been included in the revised draft in Section 5.2.4 (page 8, top and bottom; page 9, top) and Appendix G (page 23). These changes have been added in blue font to make them easy to detect.
> >
> > Note that the architecture we used (to match the requirements of the CUB data set) gave MNIST and Fashion a higher capacity (layers of size 256 vs 64) and the ability to train the input transformation for each task individually (instead of keeping it fixed as described in Appendix E.2), explaining the higher performance of most methods in those two data sets compared to the results in Table 3.
> >
> > Table 2: Average final accuracy across tasks on the Combined data set using soft layer ordering. Each column shows accuracy on the subset of tasks from each given data set, as labeled. Standard errors after ±.
> >
> > | Base   | Algorithm         | All datasets   | MNIST         | Fashion          | CUB                |
> >
> > |:---------|:--------------------|:-------------------|:----------------|:------------------|:------------------|
> >
> > | ER       | Dyn. + Comp.   | **86.5±1.8**%      | **99.5±0.0**%    | **98.0±0.3**%     | **74.2±2.0**%     |
> >
> > | ER       | Compositional | 82.1±2.5%      | **99.5±0.0**%    | 97.8±0.3%     | 65.5±2.4%     |
> >
> > | ER       | Joint                   | 72.8±4.1%      | 98.9±0.3%    | 97.0±0.7%     | 47.6±6.2%     |
> >
> > | ER       | No Comp.         | 47.4±4.5%      | 91.8±1.3%    | 83.5±2.5%     |  7.1±0.4%       |
> >
> > |:---------|:--------------------|:------------------|:------------------|:-----------------|:------------------|
> >
> > | EWC   | Dyn. + Comp.   | **75.1±3.2**%      | 98.7±0.5%     | **97.1±0.7**%     | **52.4±2.9**%     |
> >
> > | EWC   | Compositional | 71.3±4.0%      | **99.4±0.0**%     | 96.1±0.9%     | 44.8±3.5%     |
> >
> > | EWC   | Joint                   | 52.2±5.0%      | 85.1±5.5%     | 88.6±3.8%     | 17.5±1.5%     |
> >
> > | EWC   | No Comp.         | 28.9±2.8%      | 52.9±1.6%     | 52.5±1.4%     |  5.0±0.4%       |
> >
> > |:---------|:--------------------|:------------------|:------------------|:-----------------|:------------------|
> >
> > | VAN   | Dyn. + Comp.   | **75.5±3.2**%      | **99.1±0.3**%     | **96.2±0.9**%     | **53.3±2.8**%     |
> >
> > | VAN   | Compositional | 70.6±3.8%      | 98.5±0.5%     | 95.6±0.8%     | 44.2±3.5%     |
> >
> > | VAN   | Joint                   | 52.7±4.9%      | 85.5±4.9%     | 88.5±3.7%     | 18.4±1.7%     |
> >
> > | VAN   | No Comp.         | 34.6±3.7%      | 61.3±3.8%     | 59.8±3.6%     |  8.7±0.5%       |
> >
> > |:---------|:--------------------|:------------------|:------------------|:-----------------|:------------------|
> >
> > | FM     | Dyn. + Comp.   | **83.8±2.0**%      | **99.6±0.0**%     | **98.3±0.3**%     | **68.7±1.5**%     |
> >
> > | FM     | Compositional | 74.6±3.1%      | 99.5±0.0%     | 98.1±0.3%     | 50.3±2.0%     |

---

> > > ### Comment · AnonReviewer2 · 2020-11-23
> > > **Intersting new results**
> > >
> > > Thank you for the additional results, especially the experiment on the diverse stream of tasks clearly showing that the compositional approach is indeed able to outperform more rigid strategies.
> > > I updated my score accordingly.

---

### Official Review · AnonReviewer5 · 2020-11-06
**No comparison to previous work, unclear why**

**Rating:** 6
**Confidence:** 3

**Review:**

### Summary
The paper attempts to solve lifelong/continual learning (CL) by building reusable components and learning both the way of combining them and the components themselves. To do this, the authors present a framework of algorithms which is based on three abstract elements which are iterated:
a. Updating the components itself
b. Updating the way the components are combined (structure) for a given task
c. Adding new components

The framework is instantiated as a concrete algorithm in several different ways whose performance is evaluated through extensive experiments.

### Concern
My biggest concern regarding the work is lack of comparison with previous CL literature. Authors do a good work listing previous contributions in CL (in Sec. 2), and provide two claims, which, as implied, would make the previous models significantly different from their work:
A: "it is unclear what the reusability of these parameters [of previous works] mean"
B: "the way in which parameters are reused is hard-coded into the architecture design"

A. Many of the previous works (in particular Achille et al. 2018 and Rao et al. 2019) contain a shared sub-model which maps data into a representation which is further processed to predict task-specific labels. The ability to extract structure from data coming from various tasks is, on the level of the objective, the same "reusable knowledge" that the authors' work attempts to gather. One could argue that the way previous work learn this knowledge could be improved: to do this, one should compare the new model with the previous ones using the existing objectives which try to assess the level of forward/backward transfer (which, arguably, is an established way of measuring "reusability of the knowledge") or by proposing a different metric for quantifying the "discovery of reusable knowledge". The work under review does neither.
B. Authors claim that hard-coded network architecture is an unsatisfactory element of previous work, in particular because it makes it harder "to learn tasks with high degree of variability". Deciding which part of the NN-based algorithm is "architecture" is fuzzy at best. I believe that the authors' work (in particular "soft layer ordering" compositional structure) can also be interpreted as learning a neural network with a fixed architecture, sharing many layers, and placing non-linearities every second layer (see * below). I would like to ask the authors to clarify what formally they mean by "hard-coding architecture". Furthermore, the hypothesis of "methods with hard-coded architecture have a disadvantage at 'hard' tasks" needs experimental verification, which I found lacking in the paper.

### Smaller statements
1. The paper is well-written and relatively easy to understand.
2. The work seems very related to the domain of Neural Architecture Search: I believe a reference to this line of work is warranted.
3. If I understood correctly, the method is trained on batches of 4 tasks (or are 4 tasks present only in the first batch?) at a time. This is a significant simplification of the classical CL setting where only one task can be observed at a time. I believe a big part of the CL problem is that one never has an access to the level of "task variability" and thus needs to trade-off overfitting and underfitting without having access to the meta-task.
4. The work is missing a clear message: the presented framework is a generalization of some previous methods, but it's far from clear why the algorithms fitting to the framework would be beneficial over the others solving the same problem of CL nor what other benefit does thinking in terms of the proposed framework gives.

### A question
why $\psi$ and $s_j$ sum to 1 in Sec. 4.1?

### Review summary
I find the overall method interesting and a potentially advantageous approach to CL. At the same time, despite extensive experiments and a detailed writeup (spilling, a bit unfairly, crucial parts of the work (like an algorithm diagram) into a 10+ page-long appendix), I feel that placing the work in the contemporary CL+NAS literature is inadequate, making it impossible to appreciate the benefits of the work.

*
Let $x \in R^d$: a data point, $\sigma$: a non-linear function, $A_{1, 2}$: activation matrix $\in R^{d \times d}$.

A typical, 2-layer network could be described as:
$$
f(x) = \sigma(A_2 \sigma (A_2 x))
$$

Your soft layer ordering model is (dropping some indices):
$$
f(x) = \sum_i \psi_{i,1} m_i \circ \sum_j \psi_{j, 2} m_j(x) =
$$
$$
= \sum_i \psi_{1, i} \sigma \Big( \phi_i^T \times \big(\sum_j \psi_{2, j} \sigma(\phi_j^T x)\big)\Big) =
$$$$
= \psi_1 \sigma(\phi \psi_2 \sigma(\phi x))))
$$
Where $\phi$ is of shape $k \times d \times d$ and $\psi_1, \psi_2 \in R^k$
In other words, the proposed model behaves like a neural network with interleaving $d$- and $k$-units-wide hidden layers, with and without non-linearities and some extra weight sharing and broadcasting.

---

> ### Author Response · Authors · 2020-11-14
> **Thanks for your detailed feedback. We have added clarifications about our baselines and compared against an additional suggested baseline (1/2)**
>
> ### Concern
> We compare against several existing approaches: EWC and ER, which are among the most popular choices of continual learning methods, and two naive baselines: fine-tuning and freezing shared parameters. We consider both non-compositional (the standard architectures used in past work) and jointly trained (the naive way to learn compositional structures with existing approaches) variants.
>
> A. _Comparison against other approaches to extracting reusable knowledge:_ As you point out, there are many works which use continual learning to extract a shared representation to leverage for future use. This is conceptually equivalent to our no-components baselines, which use a set of common layers shared across all tasks, with task-specific input and output mappings. We have added a clarification about no-components baselines to our draft (page 5, bottom). Our evaluations consistently show that compositional representations of the knowledge increase continual learning performance substantially.
>
> In particular, CURL (Rao et al., 2019) can be considered a no-components baseline, not fundamentally different from the ones in Table 3. In its supervised-learning version, CURL shares a group of layers to build a task-agnostic representation, and then uses task-specific output layers to classify each input sample. Additionally, it contains _class-and-task_-specific encoders to map into a latent space, which is later decoded via self-supervised learning to re-construct the input. This latter encoding-decoding portion is used for generative replay to avoid forgetting.
>
> To validate that our approaches indeed outperform CURL, we ran an additional evaluation on MNIST and Fashion MNIST using CURL. To match the experimental setting to ours, the shared encoder used four 64-unit layers shared across all tasks, plus task-specific classification layers ($q(\mathbf{y} \mid \mathbf{x})$). The class-and-task-specific encoders mapped the shared representation into a hidden encoding $\mathbf{z}$, which was further processed by four 64-unit shared layers to output the re-constructed input. We used the [official implementation](https://github.com/deepmind/deepmind-research/tree/master/curl), with minor modifications to consume the data sets we use in our evaluation. For these experiments, we initialized CURL with batch MTL on the first 4 tasks like our algorithms. Note also that for these experiments, CURL was given the following advantages over our methods to better match the experimental setting in Rao et al.'s (2019) work: more shared parameters (124112 vs 29120), more task-specific parameters (8450 vs 93 per task), storage of more bytes for previous tasks to avoid forgetting (834448 vs 250880 total), and no random linear projection of the input (so the network can directly observe commonalities between repeated classes across tasks). Compared to the ER baselines in the paper, CURL alternates one backpropagation step on the current task's data with one step on replayed data across all past tasks, whereas our baselines only replay each stored example once per epoch, so CURL uses substantially more computation.
>
> The table below shows the obtained accuracy values for ER compositional and ER dynamic+compositional using soft-ordering (from Table 3) and CURL, displaying a small yet  significant improvement of our approach. We note that since CURL is equivalent conceptually to our no-components baselines, we place it in the same category as those, to suggest that a compositional version of CURL _is_ possible, by adding a compositional structure (e.g., soft layer ordering) to the shared sub-network in CURL.
>
> Table 1: Average accuracy of CURL and ER compositional variants using soft layer ordering. Standard errors reported after the ±.
>
> | Base   | Algorithm         | MNIST       | Fashion      |
>
> |:---------|:--------------------|:---------------|:---------------|
>
> | ER       | Dyn. + Comp.   | **97.7±0.2**% | **96.3±0.4**% |
>
> | ER       | Compositional | 96.5±0.2% | 95.3±0.7%  |
>
> | CURL  | No Comp.         | 96.9±0.3% | 94.5±0.7%  |
>
> Regarding the metrics used in our evaluation, as pointed out by Lopez-Paz and Ranzato (2017), forward transfer is not a sensible metric when the task descriptors are integer indicators (as is the case in our experiments), since they do not enable zero-shot generalization. An equivalent metric can be computed for our compositional methods by considering the performance after the assimilation phase instead of zero-shot performance. However, this does not apply to the remaining baselines which adapt the shared parameters alongside any task-specific parameters, conflating adaptation with knowledge reuse. (continues below)

---

> > ### Author Response · Authors · 2020-11-14
> > **Thanks for your detailed feedback. We have added clarifications about our baselines and compared against an additional suggested baseline (2/2)**
> >
> > On the other hand, backward transfer _is_ applicable. While we don't show the average metric proposed by Lopez-Paz and Ranzato (2017), Figure F.4 in Appendix F shows a similar metric (a ratio instead of a difference) per task, clearly showing that our approach has a much better backward transfer than the baselines. In the table below, we report the average backward transfer, again showing that our methods indeed find better reusable knowledge.
> >
> > Table 2: Average backward transfer (Lopez-Paz and Ranzato, 2017) of CURL and ER compositional variants using soft layer ordering. Standard errors reported after the ±.
> >
> > | Base   | Algorithm         | MNIST      | Fashion    |
> >
> > |:---------|:--------------------|:--------------|:--------------|
> >
> > | ER       | Dyn. + Comp.   | -0.9±0.1% | -0.7±0.1% |
> >
> > | ER       | Compositional | **-0.5±0.1**% | **-0.2±0.2**% |
> >
> > | CURL  | No Comp.         | -2.2±0.3% | -3.0±0.4% |
> >
> > B. _Hard-coding architecture_: We agree that the soft-ordering net can be viewed as a neural net as you described. However, note that the linear layers between consecutive non-linear layers (the structure parameters, $\psi$) are _task-specific_. One way to view this is as simply interleaving task-specific and shared layers, as you correctly point out. However, another possible interpretation is that each $\phi_i$ is in charge of selecting which of the $\psi_j$'s to use at each given depth. This is exactly the case in the limit where the $\phi_i$'s are one-hot vectors, but only approximately so if they aren't. This interpretation is the same as made by other work in learning compositional structures (e.g., Meyerson and Miikkulainen, 2018; Kirsch et al., 2018). Going back to the example of soft-ordering nets, a non-compositional architecture (or hard-coded) would use the layers in a fixed, prescribed order ($\phi_1 \to \phi_2 \to \phi_3 \cdots$), whereas our architecture enables reordering the layers in the best way possible for a given task, as learned by the $\psi_i$'s. We reiterate that this is precisely the reason why we included no-components baselines in our experiments, which use fixed architectures and underperform our compositional variants substantially.
> >
> > ### Smaller statements
> >
> > 1. Thanks for the positive note about our writing.
> > 2. Thanks for pointing out the connection to NAS. We added the following note to Section 2 to point out that many of the works we cite can indeed be considered instances of NAS: "Many of these works can be viewed as instances of neural architecture search, a closely related area (Elsken et al., 2019)" (page 2, middle of page).
> > 3. We **do not** train on batches of 4 tasks, except during initialization. Tasks are trained consecutively one-by-one after initialization. We give this same 4-task initialization capability to all baselines for a fair comparison. We made this choice because it is a very simple mechanism of initializing components without suffering from any effect of forgetting, but other initialization methods could be applied. For example, one could still choose a fixed random structure for the initial tasks that reuses components like we do, but train those initial tasks following our assimilation-accommodation procedure instead of batch multi-task training.
> > 4. _Clear message:_ Many recent works have shown that compositional structures enable improved learning in the multi-task setting, and some have even pointed out the relevance of these ideas for lifelong or continual learning (e.g., Chang et al., 2019; Kirsch et al., 2018), but have failed to learn such structures in the lifelong setting. This motivated us to formulate a framework that can learn these types of compositional structures in a lifelong learning setting. Since motivation alone is not sufficient, we compare against two types of baselines. The no-components baselines are the traditional variants of existing lifelong methods (primarily EWC and ER), and our methods substantially outperform them, showing the benefits of learning compositional structures. The jointly trained baselines use the same compositional structures as our methods, but do not follow our framework and instead directly apply existing lifelong approaches. We also show that our methods substantially outperform these baselines. The take-away is then that compositional structures enable better lifelong learning than non-compositional structures, and that our framework enables learning such compositional structures. Future work building upon our proposed framework either by extending it or by studying approaches for any of the 4 steps has the potential of being very impactful to the field.
> >
> > ### A question
> >
> > _Why do structures sum to 1?:_ Hard layer ordering would require each layer's selection vector ($\psi_i$ for soft ordering and $s_i$ for soft gating) to be a one-hot vector. Restricting them to sum to one is a soft approximation to this, proposed by Meyerson and Miikkulainen (2018).

---

> > > ### Comment · AnonReviewer5 · 2020-11-17
> > > **Good job on experiments, I not fully agree with the explanations**
> > >
> > > Thank you for your detailed answer and congratulations on managing to compare to CURL in such a short time.
> > >
> > > A. I encourage you to include these evaluation results in your writeup (likely in the appendix), as, in my opinion, they are a useful contribution.
> > >
> > > B+4. Thank you for your clarification and references to the previous work: it helped to put your work in context. Maybe I missed it when reading Meyerson and Miikkulainen (2018), but I failed to find there the interpretation that "that each $\phi_i$ is in charge of selecting which of the $\psi_j$'s to use at each given depth" (is it in sec. 3.3.?). Without more experimental evidence, it's impossible to tell whether this "approximation" is anywhere close to reality. Further, I cannot agree that "Each component m is a self-contained, reusable function parameterized by $\phi_i$ that can be combined with other components.", as, in fact, the "components" are being combined in a soft way at the end of the day.
> > > Similarly, referring to Chang et al. (2019) and Kirsch et al. (2018) for a motivation on learning compositional structures is unwarranted, as these works do actually learn independent modules, as opposed to softly combining them in consecutive layers.
> > >
> > > 3. The setting in the paper is close to the classical lifelong learning, but it's not exactly the same: typically one would have access to only one task at a time (see eg. Kirkpatrick at el, 2016), including during initialization. The proposed setting which gives access to multiple tasks at a start is simpler: it's hard to measure exactly how much simpler. I agree it's easy to adapt the proposed methods to the classical lifelong learning setting, and I imagine re-running experiments in it to be a lot of work, the result of which *may* be the same.
> > >
> > > At the same time, I consider adding such simplifications to the setting without good arguments harmful to the domain: from now on, other methods attempting to solve CL will need to compare to the authors' work and may find it hard to beat their numbers without restricting to the simplified setting themselves.
> > >
> > > Overall, I appreciate the benefits of the work and I now believe the method itself passes the acceptance threshold. At the same time, I question the language the authors use, as both "compositional structures" and "lifelong learning" are used in a different way than the community does, which is misleading. I updated my score.

---

> > > > ### Author Response · Authors · 2020-11-23
> > > > **Thank you for updated feedback. We respond to your comments below.**
> > > >
> > > > A. Thank you. We are glad that you found our additional experiments valuable, and we will add those to the final revised draft.
> > > >
> > > > B+4. Meyerson and Miikkulainen (2018) followed the layer ordering interpretation to motivate their work, and as explained in Section 3.3 in their paper, their proposed model enables the agent to "learn _how_ layers are applied" and to discover "layers that are used in different ways at different depths for different tasks." They validate that their approximation gets closer to a hard ordering over time in Figure 4-c. Your point that the compositions we learn are soft is entirely true, but we contend that this does not make our approach any less compositional: modules are still being combined in different ways for different tasks. We will update our phrasing to clarify that our _motivation_ is to learn self-contained, reusable functions, but that we don't explicitly encourage this. Still, our results in Figure 3 and Appendix H suggest that our approach  finds such compositional structures. Independence or sparsity could be explicitly incorporated into our method by simply adding sparsity penalties, using a Gumbel softmax in our selection function, or directly following Kirsch et al. (2018) or Chang et al. (2019) in our assimilation stage. We chose to follow Meyerson and Miikkulainen's (2018) approach for simplicity, and showed quantitatively and qualitatively that our methods find better compositional structures than alternative training methods.
> > > >
> > > > 3\. We added the multi-task initialization stage to emphasize the importance of properly initializing the components in such a way that the later tasks' assimilation stage can succeed. Note that some of the early work on lifelong learning treated the first few tasks as a special initialization period, used differently from the remaining tasks. Our proposed initialization is the simplest way we found to achieve generalization of the modules to new tasks and layer orders, and we hope that future work will find alternatives that operate entirely online. The purpose of our work is to provide insight into how compositional structures can be learned over a sequence of tasks, which is compatible with the initialization method we propose.

---

### Author Response · Authors · 2020-11-14
**Thank you. Updated paper draft, and individual responses to each reviewer.**

Thank you to all reviewers for your valuable feedback. We appreciate the level of detail with which you analyzed our contributions. We would like to thank especially Reviewers 4 and 3 for their very high assessment of our work, noting the thoroughness of our presentation and experimental evaluation, and the potential impact of our work. We have updated our draft to address some of the concerns raised in the reviews. Primarily, we have added:
- A clarification that no-components baselines in our Tables 2, 3, and 4 are standard lifelong learning methods (page 5, bottom)
- A conclusion section to close our draft (page 9)
- A connection to neural architecture search in Section 2 (page 2, middle)
- References pointed out by Reviewer 2 in Section 2 (page 2, bottom)
- Notational symbols to the caption of Table 1 (page 5, middle)

All changes are typed in BLUE color for maximum clarity.

We have also responded to each reviewer individually to clarify any misunderstandings and answer any questions raised in the reviews.

---

> ### Author Response · Authors · 2020-11-18
> **We have updated the revised draft with additional experiments on a harder data set combining MNIST, Fashion MNIST, and CUB.**
>
> Following the suggestion of Reviewer 2, we have evaluated our methods and baselines on a harder data set combining tasks from MNIST, Fashion MNIST, and CUB. In this setting, composition is substantially more beneficial, given the diversity of the tasks. Please see our response to Reviewer 2 for more details, or refer to our updated draft in Section 5.2.4 (page 8, top and bottom; page 9, top) and Appendix G (page 23).

---

### Decision · Program_Chairs · 2021-01-07
**Final Decision**

**Decision:**

Accept (Poster)

**Comment:**

The paper addresses lifelong/continual learning (CL) by combining reusable components. The algorithm is based on, updating components, updating how they are combined for a given task and adding new components.

Reviewers had concerns about the learning workflow, how it could scale to harder CL streams and how it differs from existing LL/CL work.  They also asked for clarifications about compositionality. They highlighted the experiments as a point of strength.  After the rebuttal, all reviewers found the paper to be above the acceptance bar.